# ADVANTAGE ALIGNMENT ALGORITHMS

**Juan Duque\*, Milad Aghajohari\*, Tim Cooijmans, Razvan Ciuca, Tianyu Zhang,**
**Gauthier Gidel**, **Aaron Courville**
University of Montreal & Mila
`firstname.lastname`@umontreal.ca

## ABSTRACT

Artificially intelligent agents are increasingly being integrated into human decision-making: from large language model (LLM) assistants to autonomous vehicles. These systems often optimize their individual objective, leading to conflicts, particularly in general-sum games where naive reinforcement learning agents empirically converge to Pareto-suboptimal Nash equilibria. To address this issue, opponent shaping has emerged as a paradigm for finding socially beneficial equilibria in general-sum games. In this work, we introduce Advantage Alignment, a family of algorithms derived from first principles that perform opponent shaping efficiently and intuitively. We achieve this by aligning the advantages of interacting agents, increasing the probability of mutually beneficial actions when their interaction has been positive. We prove that existing opponent shaping methods implicitly perform Advantage Alignment. Compared to these methods, Advantage Alignment simplifies the mathematical formulation of opponent shaping, reduces the computational burden and extends to continuous action domains. We demonstrate the effectiveness of our algorithms across a range of social dilemmas, achieving state-of-the-art cooperation and robustness against exploitation.

## 1 INTRODUCTION

Recent advancements in artificial intelligence, such as language models like GPT (Radford et al., 2018), image synthesis with diffusion models (Ho et al., 2020), and generalist agents like Gato (Reed et al., 2022), suggest a future where AI systems seamlessly integrate into everyday human decision-making. While these systems often optimize for the goals of their individual users, this can lead to conflicts, especially in tasks that involve both cooperative and competitive elements. *Social dilemmas*, as introduced by Rapoport and Chammah (1965), describe scenarios where agents acting selfishly achieve worse outcomes than if they had cooperated. A global example is the climate change problem, where individual and national interests in economic growth often clash with the need for collective action to reduce carbon emissions and mitigate environmental degradation. The challenges we have faced in tackling this problem highlight the complexity of aligning individual interests with collective well-being.

As artificially intelligent systems become ubiquitous, there is a pressing need to develop methods that enable agents to autonomously align their interests with one another. Despite this, the deep reinforcement learning community has traditionally focused on fully cooperative or fully competitive settings, often neglecting the nuances of social dilemmas. Sandholm and Crites (1996) empirically demonstrated that naive reinforcement learning algorithms tend to converge to the worst Pareto suboptimal Nash equilibria of Always Defect in social dilemmas like the Iterated Prisoner's Dilemma (IPD). Foerster et al. (2018b), demonstrated that the same is true for policy gradient methods, and introduced *opponent shaping* (LOLA) to address this gap.

LOLA is an opponent shaping algorithm that influences the behavior of other agents by assuming they are naive learners and taking gradients with respect to simulated parameter updates. Following this approach, other opponent shaping algorithms that compute gradients with respect to simulated parameter updates have shown success in partially competitive tasks, including SOS (Letcher et al., 2021), COLA (Willi et al., 2022), and POLA (Zhao et al., 2022). More recently, LOQA (Aghajohari et al., 2024b) proposed an alternative form of opponent shaping by assuming control over the value function of other agents via REINFORCE estimators (Williams, 1992). This new approach to

opponent shaping offers significant computational advantages over previous methods and lays the foundation for our work.

We introduce *Advantage Alignment*, a family of algorithms designed to shape rational opponents by aligning their advantages when their historic interactions have been positive. We make two key assumptions about reinforcement learning agents: (1) they aim to maximize their own expected return, and (2) they take actions proportionally to this expected return. Under these assumptions, we demonstrate that opponent shaping reduces to aligning the advantages of different players and increasing the log probability of an action proportionally to their alignment. We show that this mechanism lies at the heart of existing opponent shaping algorithms, including LOLA and LOQA. By distilling this objective, Advantage Alignment agents can shape opponents without relying on imagined parameter updates (as in LOLA and SOS) or stochastic gradient estimation that relies on automatic differentiation introduced in DiCE (Foerster et al., 2018a) (as in POLA, COLA, and LOQA). Furthermore, we demonstrate that Advantage Alignment preserves Nash Equilibria, ensuring that our algorithms maintain stable strategic outcomes.

We also introduce *Proximal Advantage Alignment*, which formulates Advantage Alignment as a modification to the advantage function used in policy gradient updates. By integrating this modified advantage into the Proximal Policy Optimization (PPO) (Schulman et al., 2017b) surrogate objective, we develop a scalable and efficient opponent shaping algorithm suitable for more complex environments. To identify and overcome challenges that arise from scale—which are often overlooked in simpler settings like the Iterated Prisoner's Dilemma (Rapoport and Chammah, 1965) and the Coin Game (Foerster et al., 2018b)—we apply Advantage Alignment to a continuous variant of the Negotiation Game (Cao et al., 2018) and Melting Pot's Commons Harvest Open (Agapiou et al., 2023). In doing so, we aim to demonstrate the scalability of our methods and offer insights and solutions applicable to complex, real-world agent interactions.

Our key contributions are:

- We introduce Advantage Alignment and Proximal Advantage Alignment (PAA), two opponent shaping algorithms derived from first principles and based on policy gradient estimators.

- We prove that LOLA (and its variations) and LOQA implicitly perform Advantage Alignment through different mechanisms.

- We extend REINFORCE-based opponent shaping to continuous action environments and achieve state-of-the-art results in a continuous action variant of the Negotiation Game (Cao et al., 2018).

- We apply PAA to the Commons Harvest Open environment in Melting Pot 2.0 (Agapiou et al., 2023), a high dimensional version of the *tragedy of the commons* social dilemma, achieving state-of-the-art results and showcasing the scalability and effectiveness of our methods.

## 2 BACKGROUND

### 2.1 SOCIAL DILEMMAS

Social dilemmas describe situations in which selfish behavior leads to sub-optimal collective outcomes. Such dilemmas are often formalized as normal form games and constitute a subset of general-sum games. A classical example of a social dilemma is the Iterated Prisoner's Dilemma (IPD) (Rapoport and Chammah, 1965), in which two players can choose one of two actions: cooperate or defect. In the one-step version of the game, the dilemma occurs because defecting is a *dominant* strategy, i.e., independently of what the opponent plays the agent is better off playing defect. However, by the reward structure of the game, both the agent and the opponent would achieve a higher utility if they played cooperate simultaneously. Beyond the IPD, other social dilemmas have been extensively studied in the literature, including the Chicken Game and the Coin Game (Lerer and Peysakhovich, 2018), the latter of which has a similar reward structure to IPD but takes place in a grid world. In this paper we introduce a variation of the Negotiation Game (also known as the Exchange Game) (DeVault et al., 2015; Lewis et al., 2017), with a strong social dilemma component. Additionally, we evaluate our method on the Commons Harvest Open environment in Melting Pot 2.0 (Agapiou et al., 2023), which exemplifies a large-scale social dilemma. In this environment, agents must balance short-term personal gains from overharvesting common resources against the long-term collective benefit of sustainable use.

## 2.2 MARKOV GAMES

In this work, we consider fully observable, general sum, $n$-player Markov Games (Shapley, 1953) which are represented by a tuple: $\mathcal{M} = (N, \mathcal{S}, \mathcal{A}, P, \mathcal{R}, \gamma)$. Here $\mathcal{S}$ is the state space, $\mathcal{A} := \mathcal{A}^1 \times \ldots \times \mathcal{A}^n$, is the joint action space for all players, $P : \mathcal{S} \times \mathcal{A} \to \Delta(\mathcal{S})$ maps from every state and joint action to a probability distribution over states, $\mathcal{R} = \{r^1, \ldots, r^n\}$ is the set of reward functions where each $r^i : \mathcal{S} \times \mathcal{A} \to \mathbb{R}$ maps every state and joint action to a scalar return and $\gamma \in [0, 1]$ is the discount factor.

## 2.3 REINFORCEMENT LEARNING

Consider two agents playing a Markov Game, 1 (agent) and 2 (opponent), with policies $\pi^1$ and $\pi^2$, parameterized by $\theta_1$ and $\theta_2$ respectively. We follow the notation of Agarwal et al. (2021), let $\tau$ denote a trajectory with initial state distribution $\mu$ and (unconditional) distribution given by:

$$\mathrm{Pr}_\mu^{\pi^1, \pi^2}(\tau) = \mu(s_0)\pi^1(a_0|s_0)\pi^2(b_0|s_0)P(s_1|s_0, a_0, b_0)\ldots \tag{1}$$

Where $P(\cdot|s, a, b)$, often referred as the transition dynamics, is a probability distribution over the next states conditioned on the current state being $s$, agent taking action $a$ and opponent taking action $b$. Value-based methods like Q-learning (Watkins and Dayan, 1992) and SARSA (Rummery and Niranjan, 1994) learn an estimate of the discounted reward using the Bellman equation:

$$Q^1(s_t, a_t, b_t) = r^1(s_t, a_t, b_t) + \gamma \cdot \mathbb{E}_{s_{t+1}}\left[V^1(s_{t+1})|s_t, a_t, b_t\right]. \tag{2}$$

In policy optimization, both players aim to maximize their expected discounted return by performing gradient ascent with a REINFORCE estimator (Williams, 1992) of the form:

$$\nabla_{\theta_1} V^1(\mu) = \mathbb{E}_{\tau \sim \mathrm{Pr}_\mu^{\pi^1, \pi^2}}\left[\sum_{t=0}^{\infty} \gamma^t A^1(s_t, a_t, b_t)\nabla_{\theta_1} \log \pi^1(a_t|s_t)\right]. \tag{3}$$

Here $A^1(s, a, b) := Q^1(s, a, b) - V^1(s)$ denotes the advantage of the agent taking action $a$ in state $s$ while the opponent takes action $b$.

## 3 OPPONENT SHAPING

Opponent shaping, first introduced in LOLA (Foerster et al., 2018b), is a paradigm that assumes the learning dynamics of other players can be controlled via some mechanism to incentivize desired behaviors. LOLA and its variants assume that the opponent is a naive learner, i.e. an agent that performs gradient ascent on their value function, and differentiate through an imagined naive update of the opponent in order to shape it.

LOQA (Aghajohari et al., 2024b) performs opponent shaping by controlling the Q-values of the opponent for different actions assuming that the opponent's policy is a softmax over these Q-values:

$$\hat{\pi}^2(b_t|s_t) := \frac{\exp Q^2(s_t, b_t)}{\sum_b \exp Q^2(s_t, b)}, \tag{4}$$

where $Q^2(s_t, b_t) := \mathbb{E}_{a \sim \pi^1}[Q^2(s_t, a, b_t)]$. The key idea is that these Q-values depend on $\pi^1$, and hence the opponent policy $\hat{\pi}^2$ can be differentiated w.r.t. $\theta_1$:[1]

$$\nabla_{\theta_1} \hat{\pi}^2(b_t|s_t) = \hat{\pi}^2(b_t|s_t)\left(\nabla_{\theta_1} Q^2(s_t, b_t) - \sum_b \hat{\pi}^2(b|s_t)\nabla_{\theta_1} Q^2(s_t, b)\right). \tag{5}$$

This dependency of $\pi^2$ on $\theta_1$ leads to the emergence of an extra term in the policy gradient:

$$\nabla_{\theta_1} V^1(\mu) = \mathbb{E}_{\tau \sim \mathrm{Pr}_\mu^{\pi^1, \pi^2}}\left[\sum_{t=0}^{\infty} \gamma^t A^1(s_t, a_t, b_t)\left(\underbrace{\nabla_{\theta_1} \log \pi^1(a_t|s_t)}_{\text{policy gradient term}} + \underbrace{\nabla_{\theta_1} \log \hat{\pi}^2(b_t|s_t)}_{\text{opponent shaping term}}\right)\right]. \tag{6}$$

Aghajohari et al. (2024b) demonstrate an effective way to account for this dependency using REINFORCE. The present work builds on the ideas of LOQA, but reduces opponent shaping to its bare components to derive Advantage Alignment from first principles.

---

[1]See Appendix A.3 for a derivation of this expression.

---

**Algorithm 1** Advantage Alignment

---

**Initialize:** Discount factor $\gamma$, agent Q-value parameters $\phi^1$, t Q-value parameters $\phi_t^1$, actor parameters $\theta^1$, opponent Q-value parameters $\phi^2$, t Q-value parameters $\phi_t^2$, actor parameters $\theta^2$
**for** iteration$= 1, 2, \ldots$ **do**
    Run policies $\pi^1$ and $\pi^2$ for $T$ timesteps in environment and collect trajectory $\tau$
    Compute agent critic loss $L_C^1$ using the TD error with $r^1$ and $V^1$
    Compute opponent critic loss $L_C^2$ using the TD error with $r^2$ and $V^2$
    Optimize $L_C^1$ w.r.t. $\phi^1$ and $L_C^2$ w.r.t. $\phi^2$ with optimizer of choice
    Compute generalized advantage estimates $\{A_1^1, \ldots, A_T^1\}$, $\{A_1^2, \ldots, A_T^2\}$
    Compute agent actor loss, $L_a^1$, summing equation 3 and equation 8
    Compute opponent actor loss, $L_a^2$, summing equation 3 and equation 8
    Optimize $L_a^1$ w.r.t. $\theta^1$ and $L_a^2$ w.r.t. $\theta^2$ with optimizer of choice

---

## 4 ADVANTAGE ALIGNMENT

### 4.1 METHOD DESCRIPTION

Motivated by the goal of scaling opponent shaping algorithms to more diverse and complex scenarios, we derive a simple and intuitive objective for efficient opponent shaping. We begin from the assumptions that agents are learning to maximize their expected return, and will behave in a fashion that is proportional to this goal:

**Assumption 1.** *Each agent $i$ learns to maximize their value function:* $\max V^i(\mu)$.

**Assumption 2.** *Each opponent $i$ acts proportionally to the exponent of their action-value function:* $\pi^i(a|s) \propto \exp\left(\beta \cdot Q^i(s,a)\right)$.

Using Equation 6 and substituting $\hat{\pi}^2$ in place of $\pi^2$ (per Assumption 2), we obtain:

$$\nabla_{\theta_1} V^1(\mu) = \mathbb{E}_{\tau \sim \Pr_\mu^{\pi^1, \pi^2}} \left[ \sum_{t=0}^{\infty} \gamma^t A^1(s_t, a_t, b_t) \left( \nabla_{\theta_1} \log \pi^1(a_t|s_t) + \nabla_{\theta_1} \log \hat{\pi}^2(b_t|s_t) \right) \right].$$

The first term is the usual policy gradient. The second term is the opponent shaping term and will be our focus. Approximating the opponent's policy (right hand side of equation 4) by ignoring the contribution due to the partition function, the opponent shaping term becomes:

$$\beta \cdot \mathbb{E}_{\tau \sim \Pr_\mu^{\pi^1, \pi^2}} \left[ \sum_{t=0}^{\infty} \gamma^t A^1(s_t, a_t, b_t) \nabla_{\theta^1} Q^2(s_t, b_t) \right]. \tag{7}$$

The gradient of the Q-value can be estimated by a REINFORCE estimator, which leads to a nested expectation. Aghajohari et al. (2024b) empirically showed that this nested expectation can be efficiently estimated from a single trajectory. We take the same approach (see Appendix A.1) to obtain:

$$\beta \cdot \mathbb{E}_{\tau \sim \Pr_\mu^{\pi^1, \pi^2}} \left[ \sum_{t=0}^{\infty} \gamma^{t+1} \left( \sum_{k<t} \gamma^{t-k} A^1(s_k, a_k, b_k) \right) A^2(s_t, a_t, b_t) \nabla_{\theta^1} \log \pi^1(a_t|s_t) \right]. \tag{8}$$

The expression above captures the essence of opponent shaping: an agent should align its advantages with those of its opponent in order to steer towards trajectories that are mutually beneficial. More precisely, an agent increases the probability of actions that have high product between the sum of its past advantages and the advantages of the opponent at the current time step. For implementation details of the Advantage Alignment formula see Appendix A.6. Equation 8 depends only on the log probabilities of the agent, which allows us to create a proximal surrogate objective that closely follows the PPO (Schulman et al., 2017b) formulation:

$$\mathbb{E}_{\tau \sim \Pr_\mu^{\pi^1, \pi^2}} \left[ \min\left\{ r_n(\theta_1) A^*(s_t, a_t, b_t),\ \text{clip}\left(r_n(\theta_1); 1 - \epsilon, 1 + \epsilon\right) A^*(s_t, a_t, b_t) \right\} \right], \tag{9}$$

where $r_n$ denotes the ratio of the policy (new) after $n$ updates of the original policy (old), and:

$$A^*(s_t, a_t, b_t) = \left( A^1(s_t, a_t, b_t) + \beta\gamma \cdot \left( \sum_{k<t} \gamma^{t-k} A^1(s_k, a_k, b_k) \right) A^2(s_t, a_t, b_t) \right). \tag{10}$$

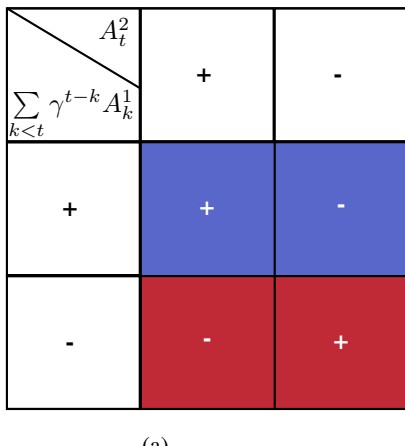 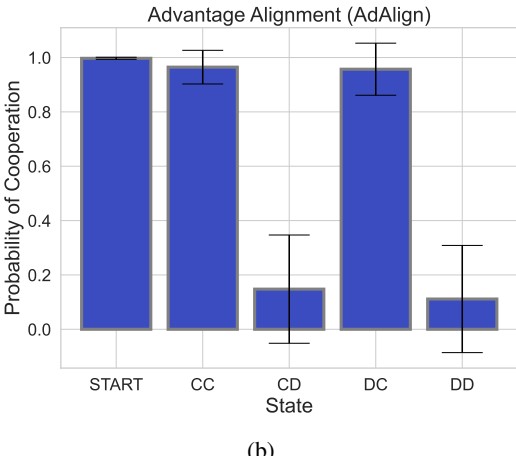

|          | (a)      |          |     (b)   |
|----------|----------|----------|-----------|

Figure 1: (a) The sign of the product of the gamma-discounted past advantages for the agent, and the current advantage of the opponent, indicates whether the probability of taking an action should increase or decrease. (b) The empirical probability of cooperation of Advantage Alignment for each previous combination of actions in the one step history Iterated Prisoner's Dilemma, closely resembles tit-for-tat. Results are averaged over 10 random seeds, the black whiskers show one std.

This surrogate objective in equation 9 is used to formulate the Proximal Advantage Alignment (Proximal AdAlign) algorithm (see appendix A.7 for implementation details).

**Why is Assumption 1 necessary?** Assumption 1 allows the agent to influence the learning dynamics of the opponent, by controlling the values for different actions. After one iteration of the algorithm the agent changes the Q-values of the opponent for different actions and, since the opponent aims to maximize their expected return, it must change its behavior accordingly.

## 4.2 Analyzing Advantage Alignment

Equation 8 yields four possible different cases for controlling the direction of the gradient of the log probability of the policy. As with the usual policy gradient estimator, the sign multiplying the log probability indicates whether the probability of taking an action should increase or decrease. Intuitively, when the interaction with the opponent has been positive (blue in figure 1a) the advantages of the agent align with that of the opponent: the advantage alignment term increases the log probability of taking an action if the advantage of the opponent is positive and decreases it if it is negative. In contrast, if the interaction has been negative (red in figure 1a) the advantages are at odds with each other: the advantage alignment term decreases the log probability of taking an action if the advantage of the opponent is positive and increases it if it is negative. We now relate existing opponent shaping algorithms to advantage alignment, and argue that these algorithms use the same underlying mechanisms. Theorem 1 shows that LOLA update from Foerster et al. (2018b) can be written as a policy gradient method with an opponent shaping term similar to equation 10. This shows the fundamental relationship between opponent-shaping dynamics and advantage multiplications. Theorem 2 proves that LOQA's opponent shaping term has the same form as that of Advantage Alignment, differing only by a scalar term.

**Theorem 1** (LOLA as an advantage alignment estimator). *Given a two-player game where players 1 and 2 have respective policies $\pi^1(a|s)$ and $\pi^2(b|s)$, where each policy is parametrised such that the set of gradients $\nabla_{\theta_2} \log \pi^2(a|s)$ for all pairs $(a, s)$ form an orthonormal basis, the LOLA update for the first player correspond to a reinforce update with the following opponent shaping term*

$$\beta \cdot \mathbb{E}_{\tau \sim \mathrm{Pr}_\mu^{\pi^1, \pi^2}} \left[ \sum_{t=0}^{\infty} \gamma^t \left( \sum_{k=t}^{\infty} d_{\gamma, k-t} \gamma^{k-t} A_k^1 A_{k-t}^2 \right) \nabla_{\theta^1} \log \pi^1(a_t|s_t) \right], \qquad (11)$$

*where $A_k^i := A^i(s_k, a_k, b_k)$ and $d_{\gamma, k}$ is the occupancy measure of the tuple $(a_k, b_k, s_k)$ and $\beta$ is the step size of the naive learner. See appendix A.2 for a proof.*

**Theorem 2** (LOQA as an advantage alignment estimator). *Under Assumption 2, the opponent shaping term in LOQA is equivalent to the opponent shaping term in Equation 8 up to $(1 - \tilde{\pi}^2(b_k|s_k))$*

$$\beta \cdot \mathbb{E}_{\tau \sim \mathrm{Pr}_\mu^{\pi^1, \pi^2}} \left[ \sum_{t=0}^{\infty} \gamma^{t+1} \left( \sum_{k<t}^{\infty} \gamma^{t-k} (1 - \tilde{\pi}^2(b_k|s_k)) A_k^1 \right) A_t^2 \nabla_{\theta^1} \log \pi^1(a_t|s_t) \right], \quad (12)$$

$\tilde{\pi}^2(b_t|s_t)$ *approximates the opponent policy as defined in LOQA. For a proof see appendix A.5.*

Having established the connection between existing opponent shaping algorithms and Advantage Alignment, we now focus on analyzing the theoretical properties of Advantage Alignment itself. We investigate the impact of the Advantage Alignment term on Nash equilibria. Theorem 3 demonstrates that Advantage Alignment preserves Nash equilibria, ensuring that if agents are already playing equilibrium strategies, the Advantage Alignment updates will not cause the policy gradient to deviate from them locally.

**Theorem 3** (Advantage Alignment preserves Nash equilibria). *Advantage Alignment preserves Nash equilibria. That is, if a joint policy $(\pi_1^*, \pi_2^*)$ constitutes a Nash equilibrium, then applying the Advantage Alignment formula will not change the policy, as the gradient contribution of the advantage alignment term is zero. The proof can be found in Appendix A.8.*

## 5 EXPERIMENTS

We follow the evaluation protocol of LOQA (Aghajohari et al., 2024b), where the fixed policy that is generated by the algorithm is evaluated *zero-shot* against a distribution of policies.

### 5.1 ITERATED PRISONER'S DILEMMA

We consider the *full history* version of IPD, where a gated recurrent unit (GRU) policy conditions on the full trajectory of observations before sampling an action. In this experiment we follow the architecture used in POLA (Zhao et al., 2022) (for details see appendix B.1). We also consider trajectories of length 16 with a discount factor, $\gamma$, of 0.9. As shown in figure 1b, Advantage Alignment agents consistently achieve a policy that resembles *tit-for-tat* (Rapoport and Chammah, 1965) empirically. Tit-for-tat consists of cooperating on the first move and then mimicking the opponent's previous move in subsequent rounds.

### 5.2 COIN GAME

The Coin Game is a 3x3 grid world environment where two agents, red and blue, take turns collecting coins. During each turn, a coin of either red or blue color spawns at a random location on the grid. Agents receive a reward of +1 for collecting any coin but incur a penalty of -3 if the opponent collects a coin of their color. A Pareto-optimal strategy in the Coin Game is for each agent to collect only the coins matching their color, as this approach maximizes the total returns for both agents. Figure 2 demonstrates that Advantage Alignment agents perform similarly to LOQA agents when evaluated against a league of different policies: Advantage Alignment agents cooperate with themselves, cooperate with Always Cooperate (AC) and are not exploited by Always Defect (AD).

### 5.3 NEGOTIATION GAME

In the original Negotiation Game, two agents bargain over $n$ types of items over multiple rounds. In each round, both the quantity of items and the value each agent places on them are randomly set, but the agents only know their own values. They take turns proposing how to divide the items over a random number of turns. Agents can end the negotiation by agreeing to a proposal, and rewards are based on how well the agreement matches their private values. If they don't reach an agreement by the final turn, neither gets a reward. We modify the game first by making the values public, otherwise Advantage Alignment would have an unfair edge over PPO agents by using the opponent's value function. Secondly, we do one-shot, simultaneous negotiations instead of negotiation rounds lasting multiple iterations. Third, we modify the reward function so that every negotiation yields a reward. For a given item with agent value $v_a$, the reward of the agent $r_a$ depends on the proposal of the agent $p_a$ and the proposal of the opponent $p_o$ where $p_a, p_o \in [0, 5]$: $r_a = p_a \cdot v_a / \max(5, p_a + p_o)$.

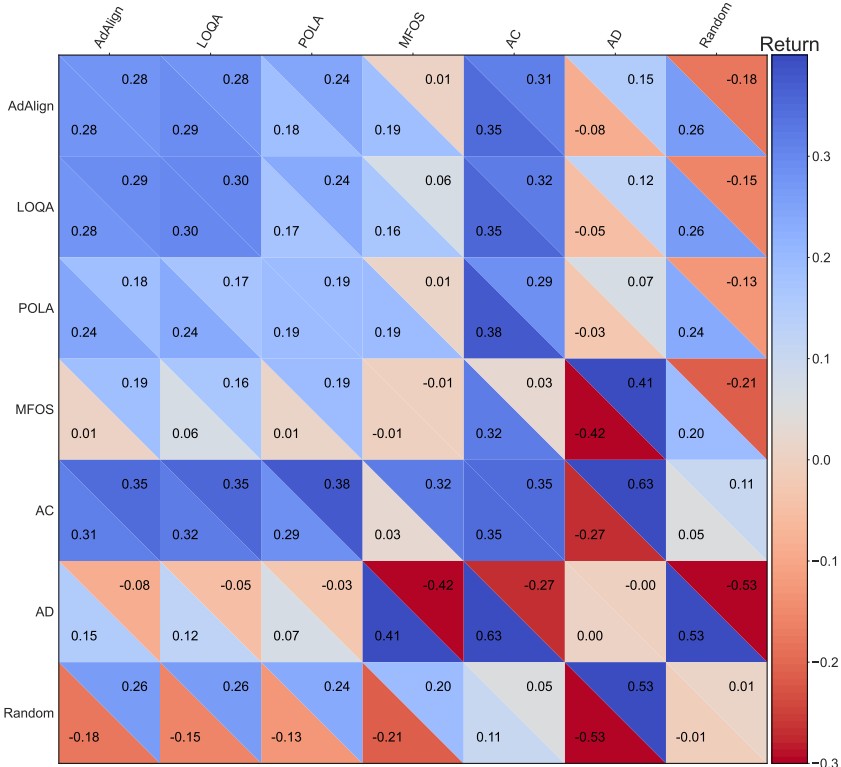

Figure 2: League Results of the Advantage Alignment agents in Coin Game: LOQA, POLA, MFOS, Always Cooperate (AC), Always Defect (AD), Random and Advantage Alignment (AdAlign). Each number in the plot is computed by running 10 random seeds of each agent head to head with 10 seeds of another for 50 episodes of length 16 and averaging the rewards.

Note that the max operation at the denominator serves to break the invariance of the game dynamics to the scale of proposals. For example, without the max operation, there would be no difference between $p_a = 1, p_o = 1$ and $p_a = 5, p_o = 5$. The social dilemma in this version of the negotiation game arises because both agents are incentivized to take as many items as possible, but by doing so, they end up with a lower return compared to the outcome they would achieve if they split the items based on their individual utilities. A Pareto-optimal strategy entails allowing the agent to take all the items that are more valuable to them, and similarly for their opponent (this constitutes the Always Cooperate (AC) strategy in Figure 3a). We experiment with a high-contrast setting where the utilities of objects for the agents are orthogonal to each other: There are two possible combinations of values in this setup: $v_a = 5, v_b = 1$ or $v_a = 1, v_b = 5$.

As shown in Figure 3a, PPO agents do not learn to solve the social dilemma. They learn the naive policy of bidding high for every item which means they get a low return against themselves. PPO agents trained with shared rewards get a high return against themselves, only to be exploited by PPO agents. They do not learn to abandon cooperation and retaliate after they are defected against. Advantage Alignment agents solves the social dilemma. They cooperate with themselves while remaining non-exploitable against Always Defect.

## 5.4 MELTING POT'S COMMONS HARVEST OPEN

In Commons Harvest Open (Agapiou et al., 2023), a group of 7 agents interact in a environment in which there is 6 bushes with different amounts of apples. Agents receive a reward of 1 for any apple consumed. Consumed apples regrow with a probability dependent on the number of apples in their $L_2$ neighborhood; specifically, if there are no apples nearby, consumed apples do not regrow. This mechanism creates a tension between agents: they must exercise restraint to prevent extinction while also feeling compelled to consume quickly out of fear that others may over-harvest.

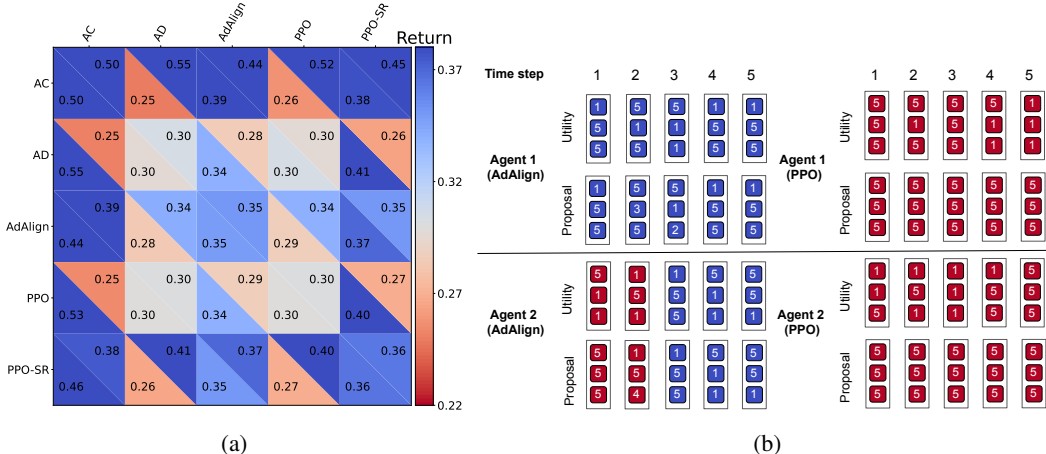

(a)

(b)

Figure 3: (a) League Results of the Advantage Alignment agents in the Negotiation Game: Always Cooperate (AC), an agent which proposes 5 for items which are more valuable to it and 1 for items that are less valuable to it, Always Defect (AD), an agent that proposes 5 regardless of the values, Advantage Alignment (AdAlign), PPO and PPO summing rewards (PPO-SR). Each number in the plot is computed by running 10 random seeds of each agent head to head with 10 seeds of another for 50 episodes of length 16 and averaging the rewards. Note that against Always Defect, Always Cooperate gets an average return of $0.25$ while Always Defect gets $0.30$. (b) Sample trajectories of AdAlign vs. AdAlign and PPO vs. PPO in the negotiation game. The numbers show the utilities and proposals, which have been rounded to integer values. AdAlign agents defect first (red) and progressively cooperate with each other (blue) while PPO agents Always Defect.

| | adalign | ppo | ppo_p | exploiter | acb | vmpo | opre | acb_p | opre_p | random |
|---|---|---|---|---|---|---|---|---|---|---|
| scenario_0 | 1.78 | 1.15 | 0.33 | 0.91 | 0.87 | 0.93 | 0.84 | 0.95 | 0.53 | 0.00 |
| scenario_1 | 1.48 | 0.74 | 0.45 | 0.76 | 0.80 | 0.85 | 0.77 | 0.94 | 0.52 | 0.00 |
| average | 1.63 | 0.94 | 0.39 | 0.83 | 0.83 | 0.89 | 0.81 | 0.94 | 0.52 | 0.00 |

Figure 4: Comparison of different reinforcement learning algorithms in Melting Pot's 2.0. Commons Harvest Open. The score is the focal return per capita, min-max normalized between a random agent and an exploiter baseline (ACB agent with an LSTM policy/value network) trained for $10^9$ steps. Following the protocol of the Melting Pot contest, we select the best agent out of 10 seeds and evaluate it 100 times.

There are a number of complications that make the Melting Pot environments particularly challenging. First, the environments are partially-observable: agents can only see a local window around themselves. Second, the partial observations are in the form of high-dimensional raw pixel data. Third, these environments often involve multiple agents—seven in the case of Commons Harvest Open—which increases the complexity of interactions and coordination. Therefore, agents need to remember past interactions with other agents to infer their motives and policies. All these factors, combined with the inherent social dilemma reward structure of the game, make finding policies that are optimal with respect to social *and* individual objectives a non-trivial task.

We train a GTrXL transformer (Parisotto et al., 2019) for 34k steps, with context length of 30, and compare the normalized focal return per capita of our agents against the baselines in Melting Pot 2.0: Advantage-critic baseline (acb) (Espeholt et al., 2018), V-MPO (vmpo) (Song et al., 2019), options as responses (opre) (Vezhnevets et al., 2020), and prosocial versions of opre (opre_p) and acb (acb_p) that encourage cooperation. We also compare to our own implementations of PPO (ppo) and PPO

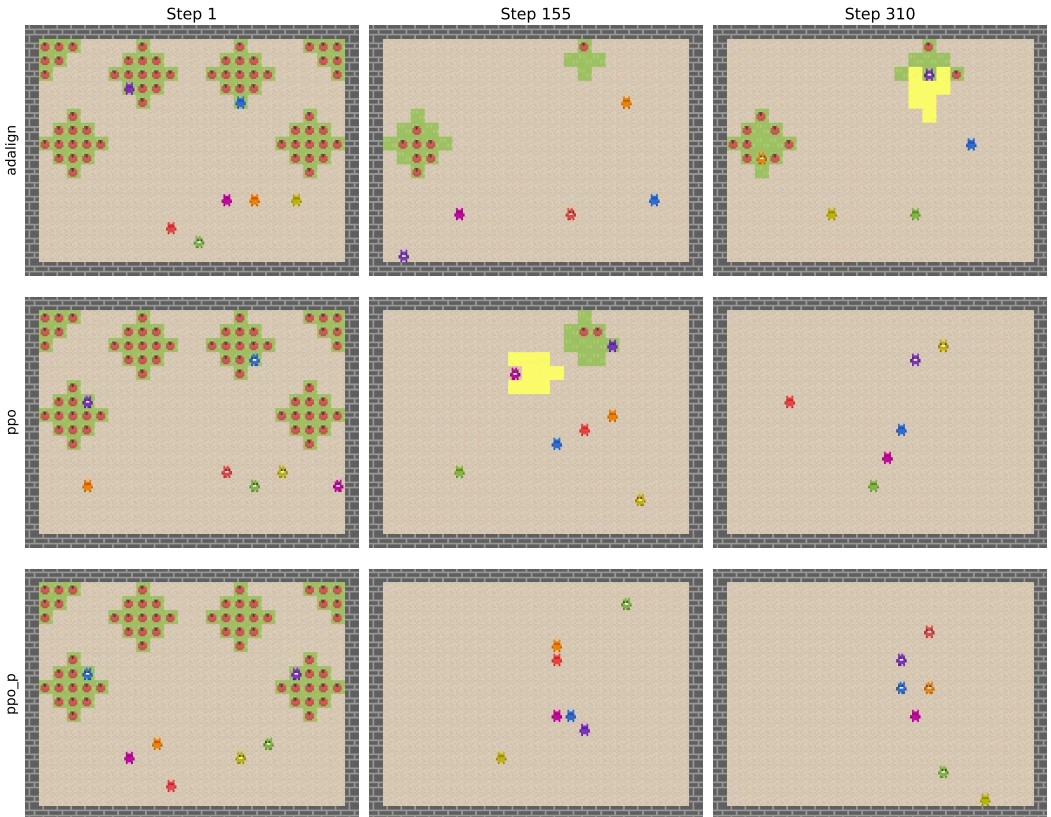

Figure 5: Frames of evaluation trajectories for different algorithms. Qualitatively, we demonstrate that Proximal Advantage Alignment (adalign) also outperforms naive PPO (ppo) and PPO with summed rewards. The evaluation trajectories show how adalign agents are able to maintain a bigger number of apple bushes from extinction (2) for a longer time that either ppo or ppo_p. Note that in the Commons Harvest evaluation two exploiter agents, green and yellow, play against a focal population of 5 copies of the evaluated algorithm.

with summed rewards (ppo_p). Figure 4 shows that our best advantage alignment agent achieves on average 1.63 normalized per capita focal return in the Commons Harvest evaluation scenarios, significantly outperforming all baselines (see Appendix B.4). Figure 5, qualitatively shows the reason why Proximal Advantage Alignment outperforms PPO and PPO with summed rewards on one of the evaluation scenarios.

# 6 RELATED WORK

The Iterated Prisoner's Dilemma (IPD) was introduced by Rapoport and Chammah (1965). *Tit-for-tat* was discovered as a robust strategy against a population of opponents in IPD by Axelrod (1984), who organized multiple IPD tournaments. It was discovered only recently that IPD contains strategies that extort rational opponents into exploitable cooperation (Press and Dyson, 2012). Sandholm and Crites (1996) were the first to demonstrate that two Q-learning agents playing IPD converge to mutual defection, which is suboptimal. Later, Foerster et al. (2018b) demonstrated that the same is true for policy gradient methods. Bertrand et al. (2023) were able to show that with optimistic initialization and *self-play*, Q-learning agents find a *Pavlov* strategy in IPD.

Opponent shaping was first introduced in LOLA Foerster et al. (2018b), as a method for controlling the learning dynamics of opponents in a game. A LOLA agent assumes the opponents are naive learners and differentiates through a one step look-ahead optimization update of the opponent. More formally, LOLA maximizes $V^1(\theta^1, \theta^2 + \Delta\theta^2)$ where $\Delta\theta^2$ is a naive learning step in the direction that maximizes the opponent's value function $V^2(\theta^1, \theta^2)$. Variations of LOLA have been introduced

to have formal stability guarantees (Letcher et al., 2021), learn consistent update functions assuming mutual opponent shaping (Willi et al., 2022) and be invariant to policy parameterization (Zhao et al., 2022). More recent work performs opponent shaping by having an agent play against a best response approximation of their policy (Aghajohari et al., 2024a). LOQA (Aghajohari et al., 2024b), on which this work is based, performs opponent shaping by controlling the Q-values of the opponent using REINFORCE (Williams, 1992) estimators.

Another approach to finding socially beneficial equilibria in general sum games relies on modeling the problem as a meta-game, where meta-rewards correspond to the returns on the inner game, meta-states correspond to joint policies of the players, and the meta-actions are updates to these policies. Al-Shedivat et al. (2018) introduce a continuous adaptation framework for multi-task learning that uses meta-learning to deal with non-stationary environments. MFOS (Lu et al., 2022) uses model-free optimization methods like PPO and genetic algorithms to optimize the meta-value of the meta-game. More recently Meta-Value Learning (Cooijmans et al., 2023) parameterizes the meta-value as a neural network and applies Q-learning to capture the future effects of changes to the inner policies. Shaper (Khan et al., 2024), scales opponent shaping to high-dimensional general-sum games with temporally extended actions and long time horizons. It does so, by simplifying MFOS and effectively capturing both intra-episode and inter-episode information.

Melting Pot 2.0 (Agapiou et al., 2023) introduces a comprehensive suite of multi-agent reinforcement learning environments that focus on social interactions and coordination challenges, providing a valuable benchmark for evaluating the scalability and effectiveness of reinforcement learning algorithms in complex, cooperative-competitive settings. The Negotiation Game, introduced by DeVault et al. (2015); Lewis et al. (2017) and subsequently refined by Cao et al. (2018), has proven to be a significant benchmark for studying general-sum games. It integrates elements of strategy and social dilemmas, necessitating that agents balance cooperation and competition to optimize their outcomes. Noukhovitch et al. (2021) analyze this complex benchmark, underscoring its importance in the field. Future investigations will turn towards an even more sophisticated simulation proposed by Zhang et al. (2022), which involves negotiations among countries and regions with diverse resource distributions and preferences in addressing climate change.

## 7 CONCLUSION

In this work, we introduced *Advantage Alignment*, a novel family of algorithms designed to address the fundamental challenge of achieving self-interested cooperation in multi-agent reinforcement learning, particularly in social dilemmas. By deriving our algorithms from first principles, we distilled opponent shaping to its core components, providing a simple yet powerful mechanism to align agents' advantages and foster mutually beneficial behaviors. Our approach unifies and generalizes existing opponent shaping methods, such as LOLA and LOQA, demonstrating that they implicitly perform Advantage Alignment through different mechanisms. This unification not only simplifies the mathematical formulation of opponent shaping but also reduces computational complexity, enabling more efficient and scalable algorithms.

Our experiments across a range of social dilemmas, including the Iterated Prisoner's Dilemma, Coin Game, and a continuous action variant of the Negotiation Game, demonstrate that Advantage Alignment consistently achieves state-of-the-art cooperation and robustness against exploitation. Notably, we extended our methods to complex, large-scale, general-sum environments like Melting Pot's Commons Harvest Open, addressing challenges that arise from partial observability, high-dimensional observations, and multi-agent interactions. In these settings, Advantage Alignment agents learned sophisticated strategies that balance individual and collective interests, showcasing the potential of our algorithms to scale to real-world applications.

The significance of our work lies in providing a principled, efficient, and scalable solution to the longstanding problem of self-interested cooperation in general-sum games. By enabling agents to autonomously align their interests with one another, Advantage Alignment paves the way for more harmonious and socially beneficial interactions in artificial intelligence systems integrated into human decision-making processes. This has profound implications for the development of AI agents in diverse domains, from autonomous vehicles navigating shared environments to AI assistants collaborating with humans and other agents.

## 8 ACKNOWLEDGEMENTS

J.D. would like to thank Michael Noukhovitch for the discussions related to the Negotiation Game. We would also like to acknowledge the technical support of Olexa Bilaniuk. This work was financially supported by A.C.'s CIFAR Canadian AI chair and Canada Research Chair in Learning Representations that Generalize Systematically.

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

# Appendix

## Table of Contents

# A MATHEMATICAL DERIVATIONS

## A.1 DERIVING THE ADVANTAGE ALIGNMENT FORMULA

In this section we derive the Advantage Alignment formula in equation equation 8 from the opponent shaping expression in equation equation 6 and assumption 2. Recall assumption 2:

$$\pi^i(a|s) \propto \exp \beta \mathbb{E}_{b \sim \pi^{3-i}(\cdot|s)}[Q^i(s,a,b)]$$

Note that if $i = 1, 3 - i = 2$ and if $i = 2, 3 - i = 1$. Recall the opponent shaping policy gradient expression:

$$\nabla_{\theta_1} V^1(\mu) = \mathbb{E}_{\tau \sim \mathrm{Pr}_\mu^{\pi^1,\pi^2}} \left[ \sum_{t=0}^{\infty} \gamma^t A^1(s_t,a_t,b_t) \left( \underbrace{\nabla_{\theta_1} \log \pi^1(a_t|s_t)}_{(A)} + \underbrace{\nabla_{\theta_1} \log \pi^2(b_t|s_t)}_{(B)} \right) \right]$$

We expand the term (B) above splitting the expectation, by assumption 2, we can write:

$$\mathbb{E}_{\tau \sim \mathrm{Pr}_\mu^{\pi^1,\pi^2}} \left[ \sum_{t=0}^{\infty} \gamma^t A^1(s_t,a_t,b_t) \nabla_{\theta_1} \log \pi^2(b_t|s_t) \right]$$

$$= \beta \cdot \mathbb{E}_{\tau \sim \mathrm{Pr}_\mu^{\pi^1,\pi^2}} \left[ \sum_{t=0}^{\infty} \gamma^t A^1(s_t,a_t,b_t) \nabla_{\theta_1} \log \exp \mathbb{E}_{a \sim \pi^1(|s_t)}[Q^2(s_t,a,b_t)] \right]$$

$$= \beta \cdot \mathbb{E}_{\tau \sim \mathrm{Pr}_\mu^{\pi^1,\pi^2}} \left[ \sum_{t=0}^{\infty} \gamma^t A^1(s_t,a_t,b_t) \nabla_{\theta_1} \mathbb{E}_{a \sim \pi^1(|s_t)}[Q^2(s_t,a,b_t)] \right].$$

where is the second line we used the fact that the expected advantage is zero. For convenience of notation we define:

$$r_t^i := r^i(s_t,a_t,b_t), \ A_t^i := A^i(s_t,a_t,b_t)$$

These are the reward and advantage of agent $i$ at time step $t$ after taking action $a_t$ and opponent taking action $b_t$. From the Bellman equation equation 2 we expand as follows:

$$\beta \cdot \mathbb{E}_{\tau \sim \mathrm{Pr}_\mu^{\pi^1,\pi^2}} \left[ \sum_{t=0}^{\infty} \gamma^t A^1(s_t,a_t,b_t) \nabla_{\theta^1} \mathbb{E}_{a \sim \pi^1(|s_t)}[Q^2(s_t,a,b_t)] \right] \tag{13}$$

$$= \beta \cdot \mathbb{E}_{\tau \sim \mathrm{Pr}_\mu^{\pi^1,\pi^2}} \left[ \sum_{t=0}^{\infty} \gamma^t A_t^1 \nabla_{\theta^1} \left( \mathbb{E}_{s'} \left[ r_t^2 + \gamma \cdot V^2(s') \Big| s_t, b_t \right] \right) \right] \tag{14}$$

$$= \beta \cdot \mathbb{E}_{\tau \sim \mathrm{Pr}_\mu^{\pi^1,\pi^2}} \left[ \sum_{t=0}^{\infty} \gamma^{t+1} A_t^1 \nabla_{\theta^1} \mathbb{E}_{s'} \left[ V^2(s') \Big| s_t, b_t \right] \right] \tag{15}$$

$$= \beta \cdot \mathbb{E}_{\tau \sim \mathrm{Pr}_\mu^{\pi^1,\pi^2}} \left[ \sum_{t=0}^{\infty} \gamma^{t+1} A_t^1 \mathbb{E}_{\tau' \sim \mathrm{Pr}_\mu^{\pi^1,\pi^2}} \left[ \sum_{k=0}^{\infty} \gamma^k A_k^2 \nabla_{\theta^1} \log \pi^1(a_k'|s_k') \Big| s_t, b_t \right] \right] \tag{16}$$

$$= \beta \cdot \mathbb{E}_{\tau \sim \mathrm{Pr}_\mu^{\pi^1,\pi^2}} \left[ \sum_{t=0}^{\infty} \mathbb{E}_{\tau' \sim \mathrm{Pr}_\mu^{\pi^1,\pi^2}} \left[ \sum_{k=0}^{\infty} \gamma^{k+t+1} A_t^1 A_k^2 \nabla_{\theta^1} \log \pi^1(a_k'|s_k') \Big| s_t, b_t \right] \right] \tag{17}$$

$$= \beta \cdot \mathbb{E}_{\tau \sim \mathrm{Pr}_\mu^{\pi^1,\pi^2}} \left[ \mathbb{E}_{\tau' \sim \mathrm{Pr}_\mu^{\pi^1,\pi^2}} \left[ \sum_{t=0}^{\infty} \sum_{k=t+1}^{\infty} \gamma^{t+k+1} A_t^1 A_k^2 \nabla_{\theta^1} \log \pi^1(a_k'|s_k') \Big| s_t, b_t \right] \right] \tag{18}$$

$$= \beta \cdot \mathbb{E}_{\tau \sim \mathrm{Pr}_\mu^{\pi^1,\pi^2}} \left[ \sum_{t=0}^{\infty} \sum_{k=t+1}^{\infty} \gamma^{t+k+1} A_t^1 A_k^2 \nabla_{\theta^1} \log \pi^1(a_k|s_k) \right]. \tag{19}$$

We use Bellman equation in line (14), and the policy gradient theorem in line (16). In line (17), we use the fact that $\gamma^t A^1(s_t,a_t,b_t)$ is measurable w.r.t. the natural filtration of the process up to time $t$, $\mathcal{F}_t$, and the independence of the two terms conditioned on $\mathcal{F}_t$ by the Markov property. In line (18) we use linearity of expectation. In line (19) we use the rule of the iterated expectations. Reorganizing the summations in causal form we get the desired result:

$$\beta \cdot \mathbb{E}_{\tau \sim \mathrm{Pr}_\mu^{\pi^1,\pi^2}} \left[ \sum_{t=0}^{\infty} \gamma^{t+1} \left( \sum_{k<t} \gamma^{t-k} A^1(s_k,a_k,b_k) \right) A^2(s_t,a_t,b_t) \nabla_{\theta^1} \log \pi^1(a_t|s_t) \right]. \tag{8}$$

A.2    PROOF OF THEOREM 1

**Lemma 1** (Policy changes under gradient ascent). *Given a policy $\pi_\theta(a|s)$ parametrised such that the set of gradients $\nabla_\theta \log \pi_\theta(a|s)$ for all pairs $(a, s)$ form an orthonormal basis, and a value function $V(\theta)$, the following holds:*

$$\frac{d}{d\alpha}\pi_{\theta+\alpha\nabla_\theta V}(a|s) = \nabla_\theta V \cdot \nabla_\theta \pi_\theta(a|s) = d_\gamma(s)\pi_\theta(a|s)^2 A(a|s) \tag{20}$$

$$d_\gamma(s) \equiv \sum_{t=0}^{\infty} \gamma^t Pr(s_t = s) \tag{21}$$

$$A(a|s) \equiv Q(a|s) - V(s) \tag{22}$$

*Proof.* The policy gradient theorem gives us:

$$\nabla_\theta V(\theta) = \mathbb{E}_{\tau \sim \pi} \sum_{t=0}^{\infty} \gamma^t A(a_t|s_t)\nabla_\theta \log \pi_\theta(a_t|s_t) \tag{23}$$

$$= \sum_{(s,a)} d_\gamma(s)\pi_\theta(a|s)A(a|s)\nabla_\theta \log \pi_\theta(a|s) \tag{24}$$

Taking the dot product of this expression with $\nabla_\theta \pi_\theta(a'|s') = \pi_\theta(a'|s')\nabla_\theta \log \pi_\theta(a'|s')$ and invoking the assumed orthonormality of gradients:

$$\nabla_\theta \pi_\theta(a'|s') \cdot \nabla_\theta V(\theta) \tag{25}$$

$$= \sum_{(s,a)} d_\gamma(s)\pi_\theta(a|s)\pi_\theta(a'|s')A(a|s)\Big(\nabla_\theta \log \pi_\theta(a|s) \cdot \nabla_\theta \log \pi_\theta(a'|s')\Big) \tag{26}$$

$$= d_\gamma(s')\pi_\theta(a'|s')^2 A(a'|s') \tag{27}$$

$\square$

**Theorem 1.** (LOLA policy gradient) *Given a two-player game where players 1 and 2 have respective policies $\pi^1(a|s)$ and $\pi^2(b|s)$, where each policy is parametrised such that the set of gradients $\nabla_{\theta_2} \log \pi^2(a|s)$ for all pairs $(a, s)$ form an orthonormal basis, the LOLA update for the first player correspond to a reinforce update with an advantage*

$$A^*_{LOLA}(s_t, a_t, b_t) = A^1(s_t, a_t, b_t) + \beta \cdot \sum_{k=t}^{\infty} d_{\gamma,k-t}\gamma^{k-t}A^1_k A^2_{k-t}. \tag{11}$$

where $A^i_k := A^i(s_k, a_k, b_k)$ and $d_{\gamma,k}$ is the occupancy measure of the tuple $(a_k, b_k, s_k)$

*Proof.* LOLA (Foerster et al., 2018b) optimizes the return of the agent under an imagined optimization step of the opponent (assuming the opponent is a naive learning algorithm). Under their notation, a LOLA agent optimizes $V^1(\theta_1, \theta_2 + \Delta\theta_2)$ where $\Delta\theta_2$ is a gradient ascent step on the parameters of the opponent $\theta_2$. Note that along this proof because we consider the method proposed by (Foerster et al., 2018b) we use their way to compute gradients. Particularly, one does not use Assumption 2, and consequently assume that $\nabla_{\theta_1} \log \pi_{\theta_2} = 0$ (and respectively $\nabla_{\theta_2} \log \pi_{\theta_1} = 0$.)

Since computing this value function explicitly is difficult, LOLA uses the first-order Taylor expansion surrogate objective:

$$V^1(\theta^1, \theta^2 + \Delta\theta_2) \approx V^1(\theta_1, \theta_2) + (\Delta\theta_2)^T \nabla_{\theta_2} V^1(\theta_1, \theta_2) \tag{28}$$

The gradient of the expression above w.r.t. the parameters $\theta_1$ of the agent is given by

$$\nabla_{\theta_1} V^1(\theta^1, \theta^2 + \alpha\nabla_{\theta_2} V^2(\theta_1, \theta_2)) = \nabla_{\theta_1} V^1(\theta_1, \theta_2) + \beta\left(\nabla_{\theta_1}\nabla_{\theta_2} V^1(\theta^1, \theta^2)\right)\nabla_{\theta_2} V^2(\theta_1, \theta_2). \tag{29}$$

The first-order terms above is computed using the Advantage form of the REINFORCE estimator, which is given by equation equation 3. Foerster et al. (2018b) derive the following REINFORCE estimator for the second-order term:

$$\nabla_{\theta_1} \nabla_{\theta_2} V^1(\theta^1, \theta^2) \tag{30}$$

$$= \mathbb{E}_{\tau \sim \mathrm{Pr}_\mu^{\pi^1, \pi^2}} \left[ \sum_{t=0}^\infty \gamma^t r_t^1 \left( \sum_{k=0}^t \nabla_{\theta_1} \log \pi^1(a_t|s_t) \right) \left( \sum_{k=0}^t \nabla_{\theta_2} \log \pi^2(b_t|s_t) \right)^\top \right] \tag{31}$$

Now, we use the following fact

$$\sum_{t=0}^\infty c_t (\sum_{k=0}^t a_k)(\sum_{l=0}^t b_l) = \sum_{t=0}^\infty c_t \sum_{S=0}^t \sum_{k=0}^S a_k b_{S-k} = \sum_{S=0}^\infty \sum_{t=S}^\infty \sum_{k=0}^S a_k b_{S-k} c_t = \sum_{S=0}^\infty \sum_{k=0}^S a_k b_{S-k} \sum_{t=S}^\infty c_t$$

to expand the second order term beginning from equation 29, to bring out the advantage $A_t^1$:

$$\nabla_{\theta_1} \nabla_{\theta_2} V^1(\theta^1, \theta^2) \tag{32}$$

$$= \mathbb{E}_{\tau \sim \mathrm{Pr}_\mu^{\pi^1, \pi^2}} \left[ \sum_{t=0}^\infty \gamma^t r_t^1 \left( \sum_{k=0}^t \nabla_{\theta_1} \log \pi^1(a_t|s_t) \right) \left( \sum_{k=0}^t \nabla_{\theta_2} \log \pi^2(b_t|s_t) \right)^\top \right] \tag{33}$$

$$= \mathbb{E}_{\tau \sim \mathrm{Pr}_\mu^{\pi^1, \pi^2}} \left[ \sum_{S=0}^\infty \left( \sum_{k=0}^S \nabla_{\theta_1} \log \pi^1(a_k|s_k) \nabla_{\theta_2} \log \pi^2(b_{S-k}|s_{S-k})^\top \right) \sum_{t=S}^\infty \gamma^t r_t^1 \right] \tag{34}$$

$$= \mathbb{E}_{\tau \sim \mathrm{Pr}_\mu^{\pi^1, \pi^2}} \left[ \sum_{t=0}^\infty \left( \sum_{k=0}^t \nabla_{\theta_1} \log \pi^1(a_k|s_k) \nabla_{\theta_2} \log \pi^2(b_{t-k}|s_{t-k})^\top \right) \sum_{l=t}^\infty \gamma^l r_l^1 \right] \tag{35}$$

$$= \mathbb{E}_{\tau \sim \mathrm{Pr}_\mu^{\pi^1, \pi^2}} \left[ \sum_{t=0}^\infty \left( \sum_{k=0}^t \nabla_{\theta_1} \log \pi^1(a_k|s_k) \nabla_{\theta_2} \log \pi^2(b_{t-k}|s_{t-k})^\top \right) \gamma^t \mathbb{E} \left[ \sum_{l=0}^\infty \gamma^l r_{l+t}^1 \right] \right] \tag{36}$$

$$= \mathbb{E}_{\tau \sim \mathrm{Pr}_\mu^{\pi^1, \pi^2}} \left[ \sum_{t=0}^\infty \left( \sum_{k=0}^t \nabla_{\theta_1} \log \pi^1(a_k|s_k) \nabla_{\theta_2} \log \pi^2(b_{t-k}|s_{t-k})^\top \right) \gamma^t Q^1(s_t, a_t, b_t) \right] \tag{37}$$

$$= \mathbb{E}_{\tau \sim \mathrm{Pr}_\mu^{\pi^1, \pi^2}} \left[ \sum_{t=0}^\infty \left( \sum_{k=0}^t \nabla_{\theta_1} \log \pi^1(a_k|s_k) \nabla_{\theta_2} \log \pi^2(b_{t-k}|s_{t-k})^\top \right) \gamma^t A_t^1 \right], \tag{38}$$

where we reorder the terms of the summation to sum over future rewards instead of past gradient terms in line 34, we use the law of iterated expectation in line 36 and a baseline subtraction in line 38.

Per Equation equation 29, we multiply this Hessian with the gradient of the value function

$$\nabla_{\theta_2} V^2(\theta_1, \theta_2) = \mathbb{E}_{\tau \sim \pi} \sum_{t=0}^\infty \gamma^t A^2(a_t, b_t|s_t) \nabla_{\theta_2} \log \pi^2(b_t|s_t) \tag{39}$$

$$= \sum_{(s,a,b)} d_\gamma(a, b, s) A^2(a, b|s) \nabla_{\theta_2} \log \pi^2(b|s) \tag{40}$$

where where $d_\gamma(a, b, s)$ is the occupancy measure of the state actions tuple $(a, b, s)$, and use the assumption that the gradients $(\nabla_{\theta_2} \log \pi^2(a|s))_{(a,s)}$ form an orthonormal basis to obtain

$$\mathbb{E}_{\tau \sim \mathrm{Pr}_\mu^{\pi^1, \pi^2}} \left[ \sum_{t=0}^\infty \left( \sum_{k=0}^t \nabla_{\theta_1} \log \pi^1(a_k|s_k) d_\gamma(a_{t-k}, b_{t-k}, s_{t-k}) A_{t-k}^2 \right) \gamma^t A_t^1 \right].$$

To completes the proof, we finally need to switch the summations to get

$$\mathbb{E}_{\tau \sim \mathrm{Pr}_\mu^{\pi^1, \pi^2}} \left[ \sum_{t=0}^\infty \left( \sum_{k=t}^\infty \gamma^k A_k^1 d_\gamma(a_{k-t}, b_{k-t}, s_{k-t}) A_{k-t}^2 \right) \nabla_{\theta_1} \log \pi^1(a_t|s_t) \right].$$

$$\square$$

## A.3 GRADIENT OF LOQA

Recall the opponent policy approximation used in LOQA, which takes a softmax over the Q-values of the opponent. We assume exact estimates of these Q-values:

$$\hat{\pi}^2(b_t|s_t) := \frac{\exp Q^2(s_t, a_t, b_t)}{\sum_b \exp Q^2(s_t, a_t, b)} \tag{4}$$

Note that $\hat{\pi}^2(b_t|s_t)$ is differentiable w.r.t. the parameters $\theta_1$ of the policy of the agent because the Q-values depend on $\pi_1$. Therefore, we can use the policy gradient theorem (see Aghajohari et al. (2024b)) to differentiate the value function of the opponent w.r.t. the parameters of the agent. For convenience of notation we define:

$$Q_t^i(b) := Q^i(s_t, a_t, b)$$

Computing the gradient of the approximated opponent's policy we get:

$$\nabla_{\theta_1}\hat{\pi}^2(b_t|s_t) = \nabla_{\theta_1} \left( \frac{\exp Q^2(s_t, a_t, b_t)}{\sum_b \exp Q^2(s_t, a_t, b)} \right) \tag{41}$$

$$= \frac{\nabla_{\theta_1} \exp Q_t^2(b_t)}{\sum_b \exp Q_t^2(b)} - \frac{\exp Q_t^2(b_t) \nabla_{\theta_1} \sum_b \exp Q_t^2(b)}{\left( \sum_b \exp Q_t^2(b) \right)^2} \tag{42}$$

$$= \frac{\exp Q_t^2(b_t) \nabla_{\theta_1} Q_t^2(b_t)}{\sum_b \exp Q_t^2(b)} - \frac{\exp Q_t^2(b_t) \sum_b \exp Q_t^2(b) \nabla_{\theta_1} Q_t^2(b)}{\left( \sum_b \exp Q_t^2(b) \right)^2} \tag{43}$$

$$= \frac{\exp Q_t^2(b_t)}{\sum_b \exp Q_t^2(b)} \left( \nabla_{\theta_1} Q_t^2(b_t) - \sum_b \frac{\exp Q_t^2(b) \nabla_{\theta_1} Q_t^2(b)}{\sum_b \exp Q_t^2(b)} \right) \tag{44}$$

$$= \hat{\pi}^2(b_t|s_t) \left( \nabla_{\theta_1} Q_t^2(b_t) - \sum_b \hat{\pi}^2(b|s_t) \nabla_{\theta_1} Q_t^2(b) \right), \tag{45}$$

where we used the quotient rule in line (42) and equation equation 4 in line(45). By the chain rule, the gradient of the log probability is:

$$\nabla_{\theta_1} \log \hat{\pi}^2(b_t|s_t) = \frac{\nabla_{\theta_1}\hat{\pi}^2(b_t|s_t)}{\hat{\pi}^2(b_t|s_t)} = \nabla_{\theta_1} Q_t^2(b_t) - \sum_b \hat{\pi}^2(b|s_t) \nabla_{\theta_1} Q_t^2(b).$$

This concludes the derivation.

## A.4 GRADIENT OF LOQA IN CONTINUOUS ACTION SPACES

We derive the gradient of the opponent's policy $\pi^2(b|s)$ with respect to the agent's parameters $\theta_1$, assuming a continuous action space.

The opponent's policy is defined as:

$$\pi^2(b|s) = \frac{\exp(Q^2(s, b))}{\int_{\mathcal{A}} \exp(Q^2(s, b'))\, db'}, \tag{46}$$

where $Q^2(s, b)$ is the Q-value of the opponent for action $b$, and $\mathcal{A}$ is the continuous action space.

Our goal is to compute the gradient of $\pi^2(b|s)$ with respect to $\theta_1$, the parameters of agent 1, which affect $Q^2(s, b)$ through interactions.

Taking the log of $\pi^2(b|s)$, we get:

$$\log \pi^2(b|s) = Q^2(s, b) - \log \left( \int_{\mathcal{A}} \exp(Q^2(s, b'))\, db' \right). \tag{47}$$

The gradient of $\log \pi^2(b|s)$ with respect to $\theta_1$ is:

$$\nabla_{\theta_1} \log \pi^2(b|s) = \nabla_{\theta_1} Q^2(s, b) - \nabla_{\theta_1} \log \left( \int_{\mathcal{A}} \exp(Q^2(s, b'))\, db' \right). \tag{48}$$

Next, we compute the gradient of the log partition function $Z(s) = \int_{\mathcal{A}} \exp(Q^2(s, b')) \, db'$:

$$\nabla_{\theta_1} \log Z(s) = \frac{\nabla_{\theta_1} Z(s)}{Z(s)} = \frac{1}{\int_{\mathcal{A}} \exp(Q^2(s, b')) \, db'} \int_{\mathcal{A}} \exp(Q^2(s, b')) \nabla_{\theta_1} Q^2(s, b') \, db', \quad (49)$$

which simplifies to:

$$\nabla_{\theta_1} \log Z(s) = \int_{\mathcal{A}} \pi^2(b'|s) \nabla_{\theta_1} Q^2(s, b') \, db'. \quad (50)$$

Now, applying the chain rule to compute the gradient of $\pi^2(b|s)$, we get:

$$\nabla_{\theta_1} \pi^2(b|s) = \pi^2(b|s) \left( \nabla_{\theta_1} Q^2(s, b) - \int_{\mathcal{A}} \pi^2(b'|s) \nabla_{\theta_1} Q^2(s, b') \, db' \right). \quad (51)$$

We are allowed to interchange the gradient and the integral by applying Leibniz's rule, which holds under the following conditions: 1. $\exp(Q^2(s, b'))$ and its gradient $\nabla_{\theta_1} \exp(Q^2(s, b'))$ are continuous, as both the exponential function and the Q-value function $Q^2(s, b')$ are smooth. 2. The integral $\int_{\mathcal{A}} \exp(Q^2(s, b')) \, db'$ converges due to the boundedness of $Q^2(s, b')$ or a rapid decay over the action space. 3. We assume $\nabla_{\theta_1} Q^2(s, b')$ is bounded, ensuring the interchange of the gradient and integral is well-defined. Thus, the final expression for the gradient of the opponent's policy is:

$$\nabla_{\theta_1} \pi^2(b|s) = \pi^2(b|s) \left( \nabla_{\theta_1} Q^2(s, b) - \int_{\mathcal{A}} \pi^2(b'|s) \nabla_{\theta_1} Q^2(s, b') \, db' \right). \quad (52)$$

The Integral above is intractable, which makes continuous action LOQA hard to scale.

### A.5 PROOF OF THEOREM 2

*Proof.* In practice, LOQA deviates from the approach discussed in Appendix A.3. Specifically, it does not differentiate through all of the Q-values, but only through that of the action $b_t$ actually observed in the sampled trajectory:

$$\tilde{\pi}^2(b_t|s_t) := \frac{\exp Q^2(s_t, a_t, b_t)}{\exp Q^2(s_t, a_t, b_t) + \underbrace{\sum_{b \neq b_t} \exp Q^2(s_t, a_t, b)}_{\text{non-differentiable}}} \quad (53)$$

This choice is made because the trajectory provides an estimate of the Q-value of each opponent action $b_t$. This estimate statistically depends on the agent's actions $a_{<t}$ and therefore can be stochastically differentiated w.r.t $\theta_1$ using REINFORCE. The other Q-values will be estimated by function approximators, which depend only implicitly on $\theta_1$ and cannot be differentiated.

Differentiating equation 53 leads to a simplified gradient:

$$\nabla_{\theta_1} \tilde{\pi}^2(b_t|s_t) = \nabla_{\theta_1} \left( \frac{\exp Q^2(s_t, a_t, b_t)}{\exp Q^2(s_t, a_t, b_t) + \sum_{b \neq b_t} \exp Q^2(s_t, a_t, b)} \right) \quad (54)$$

$$= \nabla_{\theta_1} \exp Q_t^2(b_t) \frac{\left( \exp Q_t^2(b_t) + \sum_{b \neq b_t} \exp Q_t^2(b) \right) - \exp Q_t^2(b_t)}{\left( \exp Q_t^2(b_t) + \sum_{b \neq b_t} \exp Q_t^2(b) \right)^2} \quad (55)$$

$$= \exp Q_t^2(b_t) \nabla_{\theta_1} Q_t^2(b_t) \frac{\sum_{b \neq b_t} \exp Q_t^2(b) + \exp Q_t^2(b_t) - \exp Q_t^2(b_t)}{\left( \exp Q_t^2(b_t) + \sum_{b \neq b_t} \exp Q_t^2(b) \right)^2} \quad (56)$$

$$= \tilde{\pi}^2(b_t|s_t)(1 - \tilde{\pi}^2(b_t|s_t)) \nabla_{\theta_1} Q_t^2(b_t). \quad (57)$$

By the chain rule, the gradient of the log probability is

$$\nabla_{\theta_1} \log \tilde{\pi}^2(b_t|s_t) = \frac{\nabla_{\theta_1} \tilde{\pi}^2(b_t|s_t)}{\tilde{\pi}^2(b_t|s_t)} = (1 - \tilde{\pi}^2(b_t|s_t)) \nabla_{\theta_1} Q_t^2(b_t). \quad (58)$$

The difference between LOQA and Advantage Alignment lies in the extra scaling factor $(1 - \tilde{\pi}^2(b_t|s_t))$ which accounts for the partition function. Plugging equation 58 into the generalized policy gradient equation equation 6 proves the theorem. $\square$

---

**Algorithm 2** Proximal Advantage Alignment

---

**Initialize:** Discount factor $\gamma$, agent Q-value parameters $\phi^1$, t Q-value parameters $\phi_t^1$, actor parameters $\theta^1$, opponent Q-value parameters $\phi^2$, t Q-value parameters $\phi_t^2$, actor parameters $\theta^2$
**for** iteration$= 1, 2, \ldots$ **do**
    Run policies $\pi^1$ and $\pi^2$ for $T$ timesteps in environment and collect trajectory $\tau$
    Compute agent critic loss $L_C^1$ using the TD error with $r^1$ and $V^1$
    Compute opponent critic loss $L_C^2$ using the TD error with $r^2$ and $V^2$
    Optimize $L_C^1$ w.r.t. $\phi^1$ and $L_C^2$ w.r.t. $\phi^2$ with optimizer of choice
    Optimize $L_C^1$ w.r.t. $\phi^1$ and $L_C^2$ w.r.t. $\phi^2$ with optimizer of choice
    Compute generalized advantage estimates $\{A_1^1, \ldots, A_T^1\}, \{A_1^2, \ldots, A_T^2\}$
    Compute agent actor loss, $L_a^1$, using equation 9
    Compute opponent actor loss, $L_a^2$, using equation 9
    Optimize $L_a^1$ w.r.t. $\theta^1$ and $L_a^2$ w.r.t. $\theta^2$ with optimizer of choice

---

### A.6 ADVANTAGE ALIGNMENT IMPLEMENTATION

To more clearly see the Advantage Alignment formula as an influence over each individual log probability term recall the formulation:

$$\mathbb{E}_{\tau \sim \mathrm{Pr}_\mu^{\pi^1, \pi^2}} \left[ \sum_{t=0}^{\infty} \gamma^{t+1} \left( \sum_{k<t} \gamma^{t-k} A^1(s_k, a_k, b_k) \right) A^2(s_t, a_t, b_t) \nabla_{\theta^1} \log \pi^1(a_t|s_t) \right]. \quad (8)$$

The $\gamma^t$ term helps regularize the linear scaling of the sum of the advantages of the agent. Alternatively one could regularize using $1/(1+t)$ instead:

$$\mathbb{E}_{\tau \sim \mathrm{Pr}_\mu^{\pi^1, \pi^2}} \left[ \sum_{t=0}^{\infty} \frac{1}{1+t} \left( \sum_{k<t} A^1(s_k, a_k, b_k) \right) A^2(s_t, a_t, b_t) \nabla_{\theta^1} \log \pi^1(a_t|s_t) \right]. \quad (59)$$

Which accounts to increasing the probability of the actions that align the sum of the past advantages of the agent up to the current time-step $t-1$ and the advantage of the opponent at the current time-step, $t$. In our implementation we use equation 59, as it considers more terms in the future and works better in practice.

### A.7 PROXIMAL ADVANTAGE ALIGNMENT

We can combine the two policy gradient terms into a single one to come up with a proximal Advantage Alignment formulation:

$$\mathbb{E}_{\tau \sim \mathrm{Pr}_\mu^{\pi^1, \pi^2}} \left[ \sum_{t=0}^{\infty} \gamma^t A_t^1 \nabla_{\theta^1} \log \pi^1(a_t|s_t) + \beta\gamma \sum_{t=0}^{\infty} \gamma^t \left( \sum_{k<t} \gamma^{t-k} A_k^1 \right) A_t^2 \nabla_{\theta^1} \log \pi^1(a_t|s_t) \right] \quad (60)$$

Where $\beta$ is the weight put into the Advantage Alignment loss (the negative inverse of the Boltzmann constant times the temperature). Then we have:

$$\mathbb{E}_{\tau \sim \mathrm{Pr}_\mu^{\pi^1, \pi^2}} \left[ \sum_{t=0}^{\infty} \gamma^t \left( A_t^1 + \beta\gamma \left( \sum_{k<t} \gamma^{t-k} A_k^1 \right) A_t^2 \right) \nabla_{\theta^1} \log \pi^1(a_t|s_t) \right]. \quad (61)$$

This is just the normal advantage policy gradient with a modified advantage $A^*$:

$$\mathbb{E}_{\tau \sim \mathrm{Pr}_\mu^{\pi^1, \pi^2}} \left[ \sum_{t=0}^{\infty} \gamma^t A_t^* \nabla_{\theta^1} \log \pi^1(a_t|s_t) \right], \text{ where } A_t^* = A_t^1 + \beta\gamma \left( \sum_{k<t} \gamma^{t-k} A_k^1 \right) A_t^2. \quad (62)$$

Recall the Trust Region Policy Optimization (TRPO) (Schulman et al., 2017a) objective, we want to maximize the value function while maintaining the updated policy close in policy space:

$$\max_{\theta^1} V^1(\mu)$$
$$\text{s.t. } \sup_s \left\| \pi^1(\cdot|s) - \pi_n^1(\cdot|s) \right\|_{\mathrm{tv}} \leq \delta \quad (63)$$

We can use the PPO (Schulman et al., 2017b) surrogate objective:

$$\mathbb{E}_{\tau \sim \mathrm{Pr}_\mu^{\pi^1, \pi^2}} \left[ \min \left\{ r_n(\theta_1) A_t^1, \ \mathrm{clip}\left( r_n(\theta_1); 1 - \epsilon, 1 + \epsilon \right) A_t^1 \right\} \right] \tag{64}$$

Now we apply it to the Advantage Alignment formulation that uses the modified advantage on the policy gradient equation 62:

$$\mathbb{E}_{\tau \sim \mathrm{Pr}_\mu^{\pi^1, \pi^2}} \left[ \min \left\{ r_n(\theta_1) A_t^*, \ \mathrm{clip}\left( r_n(\theta_1); 1 - \epsilon, 1 + \epsilon \right) A_t^* \right\} \right], \tag{9}$$

where we denote $\pi_n^1(a_t|s_t)$ to be the updated policy and $r_n(\theta_1) = \pi_n^1(a_t|s_t)/\pi^1(a_t|s_t)$ is the ratio between the updated policy and the old policy. We used generalized advantage estimation (GAE) (Schulman et al., 2018) to compute the advantages in this expression. Algorithm 2 summarizes the implementation of Proximal Advantage Alignment.

## A.8   PROOF OF THEOREM 3

Let $\theta_1, ..., \theta_n$ be the parameter each agent, $\pi_{\theta_i}(a|s)$ be the policies represented by those parameters, and $V_i(\theta_1, ..., \theta_n)$ be the value function of agent $i$ as a function of all the other agents.

**Lemma 2** (Zero Advantages At Nash). *For all Nash Equilibria of the game, if there exist parameters $\theta_1^*, ..., \theta_n^*$ such that $\pi_{\theta_i^*} = \pi_i^*$, where $\pi_i^*$ is the policy of agent $i$ at the Nash, then for all action-state pairs with non-zero probability under the Nash policies, we have $A_i(a|s) = 0$.*

*Proof.* By the Bellman Optimality Equation, at an optimal policy the value of agent $i$ becomes $V_i^*(s) = \arg\max_a Q^*(a, s)$, hence all actions with non-zero probability under $\pi_i^*$ have the same $Q^*(a, s)$, and since $A(a, s) \equiv Q(a, s) - V(s)$, the advantage will vanish. □

We now use lemma 2 to prove that the Advantage Alignment term is zero at a Nash equilibrium.

*Proof.* Under Advantage Alignment, the updates we take can be represented by

$$\theta_i' \leftarrow \theta_i + \alpha \cdot \mathbb{E}_{\tau \sim \mathrm{Pr}_\mu^{\pi^i, \pi^{-i}}} \left[ \sum_{t=0}^{\infty} B_t^i \, \nabla_{\theta_i} \log \pi_{\theta_i}(a_t|s_t) \right] \tag{65}$$

$$B_t^i \equiv A_t^i + \beta \cdot \mathbb{E}_{\tau \sim \mathrm{Pr}_\mu^{\pi^i, \pi^{-i}}} \left[ \sum_{j \neq i} \left( \sum_{k < t} \gamma^{t-k} A_k^i \right) A_t^j \right] \tag{66}$$

But by lemma 2, $A^j(b_t|s_t) = 0$ for all actions at a Nash, hence the second term vanishes, as does the first term for the same reason. □

## A.9   N-PLAYER ADVANTAGE ALIGNMENT

Consider the $n$-player setup, we can use the policy gradient theorem and assumption 2 to derive the following expression:

$$\nabla_{\theta_1} V^i(\mu) = \mathbb{E}_{\tau \sim \mathrm{Pr}_\mu^{\pi^i, \pi^{-i}}} \left[ \sum_{t=0}^{\infty} \gamma^t A_t^i \left( \nabla_{\theta_i} \log \pi^i(a_t^i|s_t) + \sum_{j \neq i} \nabla_{\theta_i} \log \pi^j(a_t^j|s_t) \right) \right] \tag{67}$$

Which naturally leads to the following modification to the PPO advantage following the derivation used in Proximal Advantage Alignment:

$$A_t^{i*} = A_t^i + \beta\gamma \left( \sum_{k < t} \gamma^{t-k} A_k^i \right) \sum_{j \neq i} A_t^j \tag{68}$$

Here we use the standard game theory notation of $i$ to refer to the current player and $-i$ to refer to all other players. Similarly $a_t^i$ denotes the action of player $i$ at time $t$.

# B  EXPERIMENTAL DETAILS

## B.1  ITERATED PRISONER'S DILEMMA

We use an MLP layer connected to a GRU followed by another MLP head for both the actor and critic networks, similar to the architecture used in POLA (Zhao et al., 2022). We also use a replay buffer of agents collected during training, following Aghajohari et al. (2024b). All of our IPD experiments run in 50 minutes in a nvidia A100 gpu.

Table 1: IPD Hyperparameters

| Parameter | Value |
|---|---|
| Actor Training Optimizer | Adam |
| Actor Training Entropy Beta | 0.15 |
| Actor Training Learning Rate (Actor Loss) | 0.0001 |
| Advantage Alignment Weight | 0.3 |
| Actor Hidden Size | 64 |
| Layers Before GRU | 1 |
| Q-Value Training Optimizer | Adam |
| Q-Value Training Learning Rate | 0.001 |
| Q-Value Training Target EMA Gamma | 0.99 |
| Q-Value Hidden Size | 64 |
| Batch Size | 2048 |
| Self-Play | True |
| Reward Discount Factor | 0.9 |
| Agent Replay Buffer Capacity | 10000 |
| Agent Replay Buffer Update Frequency | 1 |
| Agent Replay Buffer Current Agent Fraction | 0 |
| Advantage Alignment Discount Factor | 0.9 |

## B.2  COIN GAME

We use the same architecture used for IPD with an MLP connected to a GRU unit, followed by another MLP. We experimented with both Advantage Alignment (Equation equation 8) and Proximal Advantage Alignment (Equation equation 9), with Advantage Alignment performing better (this is the one we used). All of our Coin Game experiments run in 30 minutes in a nvidia A100 gpu.

Table 2: Coin Game Hyperparameters

| Parameter | Value |
|---|---|
| Actor Training Optimizer | Adam |
| Actor Training Entropy Beta | 0.1 |
| Actor Training Learning Rate (Actor Loss) | 0.002 |
| Advantage Alignment Weight | 0.25 |
| Actor Hidden Size | 64 |
| Layers Before GRU | 1 |
| Q-Value Training Optimizer | Adam |
| Q-Value Training Learning Rate | 0.005 |
| Q-Value Training Target EMA Gamma | 0.99 |
| Q-Value Hidden Size | 64 |
| Batch Size | 512 |
| Self-Play | True |
| Reward Discount Factor | 0.96 |
| Agent Replay Buffer Capacity | 10000 |
| Agent Replay Buffer Update Frequency | 10 |
| Agent Replay Buffer Current Agent Fraction | 0 |
| Advantage Alignment Discount Factor | 0.9 |

### B.3 Negotiation Game

We experimented with both Advantage Alignment (Equation equation 8) and Proximal Advantage Alignment (Equation equation 9), with the original Advantage Alignment performing better.

**Agent's Architecture**: The game observations are a concatenation of the availability of the items, agent's value for each item, opponent's value for each item, and previous round proposals. This makes up for an input vector of length 15. The previous round proposals are especially important as the agents need to examine whether the opponent defected against them by proposing high proposals for item in which the value of the item is higher for the agent compared to the value of the item to the opponent. In other words, if the opponent wanted to gain a little return in exchange of huge loss to the agent, defecting.

**Encoder:** The observation is then processed by an encoder. The encoder is a GRU network. The GRU network consists of first two Linear Layers with a relu non-linearity in between. Then it is passed to a GRU unit.

**Critic:** The output of the GRU is then fed to a two-layer MLP with relu non-linearities for the critic module of the agent. Additionally, we concatenate the output of the encoder with the time, the index of the step of the game, for the value function as otherwise it would be hard to estimate the value of the state without knowing how long the game is going to go on for.

**Actor:** The actor is the most complex component as it deals with continuous actions. The output of the encoder is passed to an MLP with relu non-linearities and the output of the MLP is passed to a $\tanh$ activation and scaled by $2.5$, the output of this MLP is used as the mean of a normal distribution. The logarithm of the standard deviation is modeled by a single global parameter in the actor. Next, a sample of this normal distribution is passed through a $\tanh$ activation and scaled and shifted back to $(0, 5)$. Computing the log probability of this transformations requires careful implementation. Especially if the $\mathtt{atanh}$ operation that is used is numerically unstable. Please refer to the code released with this paper for the exact implementation.

**Hyperparameters:** Please refer to 3 for the hyperparameters used in our negotiation game experiments. We use a replay buffer on our gather trajectories although the rate that it is mixed with fresh trajectories is small.

Table 3: Negotiation Game Hyperparameters

| Parameter | Value |
|---|---|
| Actor Training Optimizer | Adam |
| Trajectory Length | 50 |
| Encoder Layers | 2 |
| MLP Model Layers | 2 |
| Replay Buffer Size | 100000 |
| Replay Buffer Update Size | 500 |
| Replay Buffer Off-policy Ratio | 0.05 |
| Q-Value Training Optimizer | Adam |
| Optimizer (Actor) Learning Rate | 0.001 |
| Optimizer (Critic) Learning Rate | 0.001 |
| Entropy Beta | 0.005 |
| Advantage Alignment Weight | 3.0 |
| Self-Play | True |
| Batch Size | 16384 |
| Gradient Clipping Norm | 1.0 |

Note that the optimization of the agents in the negotiation game is unstable, preventing us from taking the last checkpoint. In our experiments in Fig 3a we select the checkpoint that corresponds to the best achieved return for the agent during the optimization of the agent and the opponent. While we are not completely certain, we observe the instability happens when the policy distribution concentrates around the maximum possible proposal which is 5. We clipped the $\mathtt{atanh}$ operation in our implementation for more numerical stability. All of our Negotiation Game experiments run in 1 hour on a nvidia A100 gpu.

## B.4 MELTING POT'S COMMONS HARVEST OPEN

We experimented with both Advantage Alignment (Equation equation 8) and Proximal Advantage Alignment (Equation equation 9), with Proximal Advantage Alignment performing better.

**Agent's Architecture**: In the Commons Harvest Open environment, agents receive observations consisting of a local view of the environment in the form of raw pixel data. Each observation is an image frame capturing the agent's immediate surroundings. We use a 3 layer convolutional neural network, following (Mnih et al., 2013), to encode the observations, which are then passed to a GTrXL tranformer (Parisotto et al., 2019).

**Encoder:** The observation frames are processed by an encoder. The encoder is a GTrXL transformer network (Parisotto et al., 2019). The GTrXL network consists of 3 transformer layers, each with a model dimension of 192 and a feedforward dimension of 192. The transformer is capable of handling sequences up to a maximum length of 1000 steps, capturing temporal dependencies in the agents' observations. In practice, we use a context length of 15.

**Critic:** The output of the encoder is then fed to a two-layer Multi-Layer Perceptron (MLP) with ReLU non-linearities for the critic module of the agent. To provide temporal context, we concatenate the current time step to the encoder's output before feeding it to the critic. This helps the critic estimate the value of the state more accurately, as the remaining time in an episode can affect the expected return.

**Actor:** The actor network shares the encoder with the critic. The output of the encoder is passed through another MLP with ReLU non-linearities to produce logits over discrete action choices. The policy is modeled as a categorical distribution over these actions, which include turning around, moving in different directions, and zapping other agents.

**Hyperparameters:** Please refer to Table 4 for the hyperparameters used in our Commons Harvest Open experiments.

Table 4: Commons Harvest Open Hyperparameters

| Parameter | Value |
|---|---|
| Self-Play | True |
| Batch Size | 512 |
| Optimizer (Actor) Learning Rate | $1 \times 10^{-5}$ |
| Optimizer (Critic) Learning Rate | $1 \times 10^{-5}$ |
| Entropy Beta | 0.1 |
| Advantage Alignment Weight | 1.0 |
| Clip Gradient Norm | 10.0 |
| Transformer Layers | 3 |
| Transformer Model Dimension | 192 |
| Transformer Feedforward Dimension | 192 |
| Discount Factor ($\gamma$) | 0.99 |
| PPO Clip Range | 0.1 |
| PPO Updates per Batch | 2 |
| Normalize Advantages | True |
| Context Length | 15 |

We use a parallelized environment with 6 copies of Commons Harvest Open to make training more efficient. Following LOQA (Aghajohari et al., 2024b), we keep a replay buffer of past agent parameters to ensure robustness against a distribution of policies. From this replay buffer we sample 2 agents at each iteration and play against 5 *self-play* agents with the current version of the policy. For each environment, we use the 5 *on-policy* trajectories to compute losses for the actor and critic. In total, our Commons Harvest Open experiments last 24 hours on an nvidia L40s gpu.

## C  ADDITIONAL FIGURES

### C.1  NEGOTIATION GAME TRAINING CURVES

Figure 6 shows the training curves of Advantage Alignment on 10 seeds.

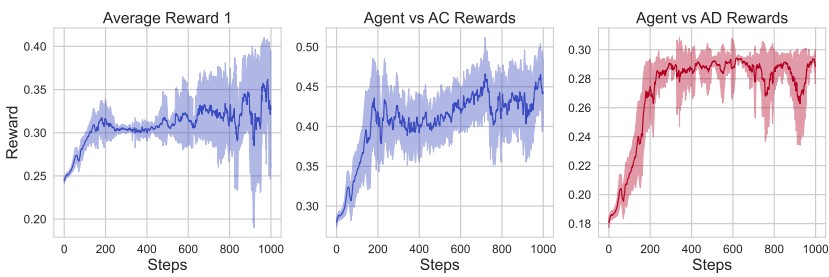

Figure 6: Training curves of Advantage Alignment averaged over 10 seeds.

### C.2  COIN GAME FULL LEAGUE RESULTS

Figure 7 shows the head-to-head results of all agents we experimented with in a league.

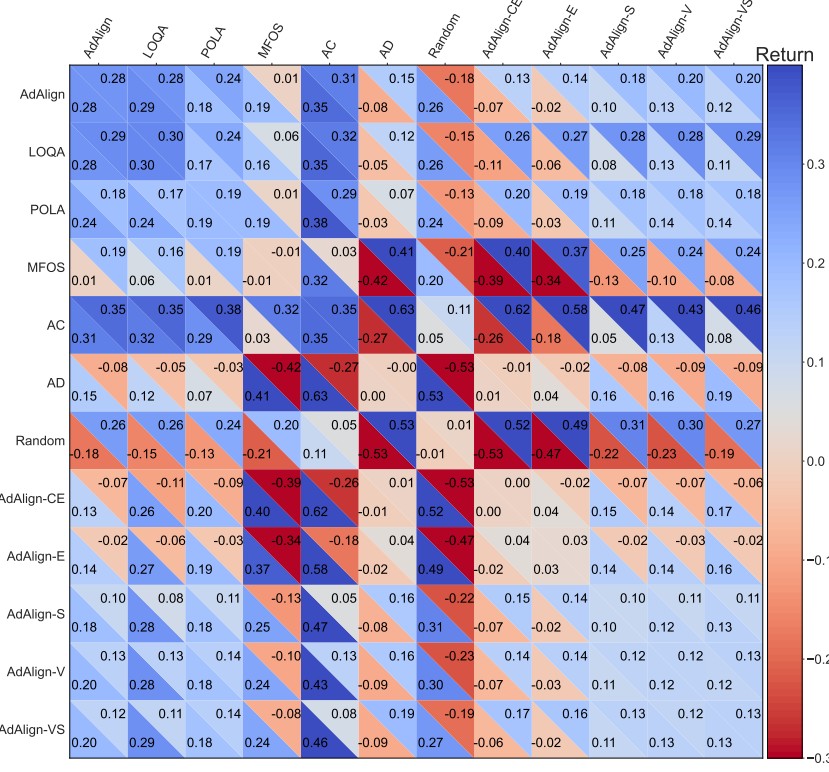

Figure 7: The head-to-head results of all variants of the coin game agents experimented with in this paper. All numbers are an average of 10 seeds of one type of agent with 10 seeds of another type of agent, where each pair play 32 games. We ablate Advantage Alignment masking different components of the gradient. Cooperative (C), masks when both advantages are positive; Empathetic (E), masks when the advantage of the agent is positive and the advantage of the opponent is negative; Vengeful (V), masks when the advantage of the agent is negative and the advantage of the opponent is positive; Spiteful (S), masks when both advantages are negative.

## C.3 ABLATION STUDY COMMONS HARVEST OPEN

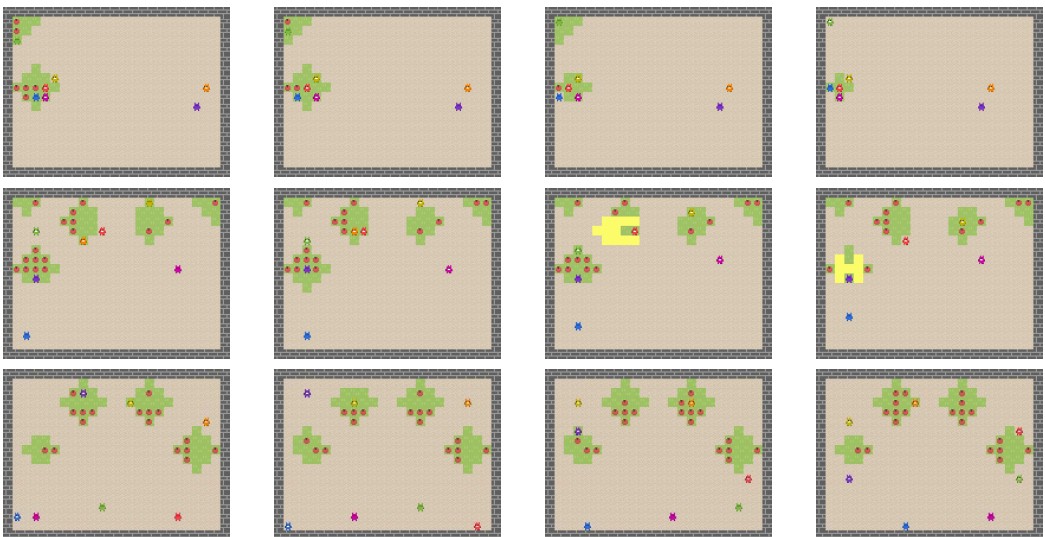

Figure 8: Sample trajectories for Proximal Advantage Alignment agents with different $\beta$ weight. We select the best of 10 seeds for each value of $\beta$. **On the first row:** $\beta = 0.5$, agents reach a policy where they try to consume the apples as fast as possible. **On the second row:** $\beta = 1$, agents reach a "bush guarding" policy, zapping any other agents coming into the same bush. **On the third row:** $\beta = 2$, agents reach a policy where they rotate around specific paths, preventing the extinction of the bushes.

Interestingly the value of $\beta$, which is used to control the weight of the advantage alignment term in equation 10, leads agents to converge to different policies. With a low value of the weight ($\beta = 0.5$), we empirically observed that most runs converge to a greedy policy that attempts to consume apples as soon as possible. With a value of $\beta = 1$, we find policies that show a "bush guarding" behavior preventing other agents from approaching their bush, and consuming apples within that bush with moderate restraint. This is the policy that shows the best evaluation performance in figure 4. With high values of the weight ($\beta = 2$), most runs find a rotating strategy in which agents stick to eating only a subset of the apples on each bush. This policy has the highest *pro-social* return out of all of them. However, the rotating strategy is also vulnerable to exploitation from greedy agents and does poorly in the evaluation scenarios. Figure 8 showcases what these policies look like in practice.

