# OpenReview forum: "Advantage Alignment Algorithms"
_ICLR.cc/2025/Conference — ICLR 2025 Oral_

### Official Review · Reviewer_V6em · 2024-10-20

**Soundness:** 3
**Presentation:** 2
**Contribution:** 3
**Rating:** 8
**Confidence:** 2

**Summary:**

The paper proposes a novel method for opponent shaping based on the advantage functions of the interacting agents. The derivation unifies prior works through the lens of advantage alignment. Empirically, the proposed method is effective in learning complex coordination behaviors under various multi-agent environments.

**Strengths:**

- The proposed derivation is novel, and provides novel insight on opponent shaping methods
- The paper is theoretically grounded and well motivated.
- The resulting derivation unifies prior works nicely (Theorem 1 and 2).
- Broad suite of baselines and evaluation environments.

**Weaknesses:**

- I wish Section 2 and 3 are more detailed, and cover more technical background on prior works. The additional information would allow the reader appreciate the contribution much better.
- The evaluation protocol is not provided. For readers that have little background in opponent shaping, it is not clear if the agent is trained prior to the evaluation or being trained (and thus adapted) during the evaluation episodes.
- It is not clear why increasing the log prob when both agents' advantages are negative is desirable (Eq. 8 and Fig. 1a).
- To my understanding, the proposed method does not outperform the baselines LOQA algorithm. The authors should discuss the differences and why it is desirable to use the proposed method over LOQA.
- The paper claims that the proposed method is more efficient than the baseline but does not elaborate on this aspect.

Minor comments:
- The advantage function is not defined mathematically.
- I believe Eq.2 is not the right expression, though I think this does not affect the main derivation of the paper.
- The connection between Eq.5 and Eq.6 is not explained.

**Questions:**

- Could the authors explain how LOLA and LOQA and AdAlign are related? I see the derivation in Theorem 1 and 2 but do not understand the implication and how they differ in terms of their training objectives.
- It is not clear why increasing the log prob when both agents' advantages are negative is desirable (Eq. 8 and Fig. 1a). Can the authors elaborate on this?

---

> ### Author Response · Authors · 2024-11-15
> **Addressing weaknesses and questions**
>
> We really appreciate the feedback provided for the reviewer and will try to address the concerns/weaknesses to the best of our knowledge:
>
> 1. We also believe that these sections might be a little short, but due to the page limit it is very difficult to do modifications to them given that we would have to compromise on other sections of the paper that we believe are equally crucial.
> 2.  This is a very valid observation and we will definitely add it to the paper, thank you. Our evaluation protocol is the same as LOQA: we train a policy using our algorithm, and deploy it in a zero-shot setting against a distribution of policies. We never do fine-tuning against other policies. Our robustness comes from the opponent shaping term and from training against past versions of the agent’s policy (agent replay buffer).
> 3. This increase in the log probabilities corresponds to retaliatory behavior. To understand why this kind of behavior is desirable, a good intuition comes from the Prisoner’s dilemma. Here, when an agent increases the probability of defection given that the other player has defected, it implicitly diminishes the Q-value of defection for the other agent. By reducing the Q-value of defection, the Q-value of cooperation becomes relatively better and agents are therefore more likely to cooperate. Similarly, in games where defection is a viable strategy, retaliation against it is desirable as it makes defection less likely to happen in the first place.
> 4. This is explored in more detail in the response to all reviewers above. A great part of our contribution is understanding, mathematically, what the gradient of LOQA is doing in the first place. LOQA has two big disadvantages with respect to Advantage Alignment, which are: lacking a PPO formulation and being intractable in continuous action spaces. We want to stress that “outperforming” is a very strong word in the multi-agent setting. This is especially nuanced here, since the performance of a policy depends on the distribution of the policies that it is evaluated against. Therefore there are almost always compromises. In the Coin Game for instance, LOQA shows higher return against itself and always defect compared to Advantage Alignment. But similarly, Advantage Alignment has a higher return against POLA and MFOS compared to LOQA. The differences are very small and both policies seem to be quite robust, which leads us to believe that we are very close to “solving” the Coin Game, in the sense of generating fixed policies that perform well against a distribution of strategies.
>
> 5. We want to clarify that Advantage Alignment has the same computational complexity as LOQA, which stressed the significant wall-clock advantages of REINFORCE based opponent shaping compared to other opponent shaping techniques including LOLA, POLA and MFOS. Additionally, Advantage Alignment has the extra empirical computational advantages of PPO.
>
> **Minor Comments:**
>
> 1. We will add the definition of the Advantage function in the paper: $A^i(s_t, a_t, b_t) := Q^i(s_t, a_t, b_t) - V^i(s_t)$.
> 2. The equation is correct, please refer to [1]. One may write an expectation over $r$ as well if we assume that the reward function is stochastic, which is not for any of our settings.
> 3. The connection lies in that, if one were to expand equation 6, one would have to write it in terms of equation 5. This is because equation 6 takes the gradient of $\log \hat{\pi}^2$ requires calculating the gradient of $\hat{\pi}^2$. But there is indeed a typo here, where we should use $\hat{\pi}^2$ in equation 6, so thank you for pointing this out.
>
> **Questions:**
>
> 1. Since this has been a point that has been brought up by other reviewers too, we delve in more detail about the similarities, differences, and advantages of Advantage Alignment compared to LOLA and LOQA in the reply to all reviewers above.
>
> 2. We have addressed this in point 3, above. But to clarify even further, this quadrant in Figure 1.a. corresponds with retaliatory behavior: it increases the probability of taking an action that hurts the opponent when the past interactions are negative.
>
> We kindly ask the reviewer to reconsider their score if we have addressed their concerns.
>
> **References**
>
> [1] Agarwal, A., Jiang, N., Kakade, S., and Sun, W. (2021). Reinforcement Learning: Theory and Algorithms. Preprint.

---

> > ### Comment · Reviewer_V6em · 2024-11-19
> >
> > Thank you the authors for providing detailed response to address my comments. I wish this additional information in the response can be included in the paper as it clarifies a lot of things in the paper. I also read the shared response and gained more understanding into the contributions of the paper. I strongly suggest the authors to rearrange the text and give more space for the conceptual explanation and further emphasis on the contributions of the work.
> >
> > As the authors have addressed all my comments, I raised my score from 5 to 8.

---

### Official Review · Reviewer_Niaw · 2024-10-31

**Soundness:** 2
**Presentation:** 3
**Contribution:** 2
**Rating:** 6
**Confidence:** 4

**Summary:**

This paper investigated a long-standing research problem in multi-agent systems and game theory, named social dilemma. To address this problem, this paper stood on the side of opponent shaping and proposed a method that attempted to align advantages of various agents, to achieve a compromise between social welfare and self-interests. The main assumption lies in the connection between the opponent's policy gradient and the agent's policy parameters. Building on this, this paper proposed a paradigm called Advantage Alignment, and a corresponding RL algorithm based on PPO. This paper also unified other approaches of opponent shaping, such as LOLA and LOQA, in the paradigm of Advantage Alignment. The experimental results on various tasks showed the effectiveness of the proposed Advantage Alignment.

**Strengths:**

1. This paper proposed an original idea that derives the advantage alignment to implement the policy gradient with respect to the opponent (the opponent shaping term), which can reduce the computational complexity. This paradigm has been shown to have connections to previous opponent shaping approaches, which is a progress in this research direction.
2. The general quality of this paper is good. Although there are some technical points that I require some furtehr clarifications, most of proofs are correct to my best knowledge. The theorem proofs are actually dependent on assumptions raised in this paper, and assumptions are provided with explanations. The experiments were conducted on sufficient banchmarks, in comparison with multiple baselines. Furthermore, the results not only include the numeric results, but also involve some demonstrations of test case, which make the paper more comprehensive.
3. This paper is well written. It has a clear motivation in Introduction and a thorough introduction of social delimma and opponent shaping. For those people who are not doing research in this direction, it is still friendly enough for them to catch up.
4. As for the significance of the research problem, social delimma, it is surely a significant problem. The main reason from my own perspective is that it can simulate a class of social problems. About the opponent shaping direction, the significance to me is sceptical, since the requirement of opponent's knowledge seems like a bit strong assumption. However, I do not deny that this is a necessary step towards the weaker and more realistic assumptions. As a result, this paper is significant within the resesrch community of opponent shaping.

**Weaknesses:**

I have some technical concerns about this paper, which are specifically listed as follows:
1. In line 720, could you explain why the $\beta$ term is missing?
2. In line 739, could you give more details about how equation (16) is derived from equation (15)?
3. In line 755, could you give more details about how to transform from (19) to (8), step by step?
4. In line 773, could you give more details about how equation (24) is derived from equation (23)?
5. In line 783, even with the assumption of orthonormal gradients, it is still not clear why equation (27) can be derived from equation (26). Could you please clarify this step by step?
6. In line 792, is $A^{*}_{\text{LOLA}}$ equal to $V^{1}$?
7. In line 809, could you explain why $\alpha \nabla_{\theta_{2}} V^{2}(\theta_{1}, \theta_{2})$ is equal to $\Delta \theta_{2}$?
8. In line 1001, there is a typo: equation equation -> equation.

Additionally, I also have some concerns about the experimental analysis:

9. In the results shown in Figure 1b, could you give some intuitive interpretation on the asymmetry between the results of CD and DC? For example, is it related to your theoretical claims?

**Questions:**

See concerns in weaknesses. If the these concerns can be resolved, I will consider to raise the score. However, for the moment I can only give a reject due to those concerns.

---

> ### Author Response · Authors · 2024-11-13
> **Addressing technical concerns**
>
> We wanted to thank the reviewer for their feedback and for their appreciation of our paper. We also believe that the quality of the paper is good. Having said that, we now address the reviewer’s technical concerns:
>
> 1. There is indeed a $\beta$ factor missing here, and we will correct this, thank you.
> 2. To proceed from equation 15 to equation 16, we simply use the policy gradient theorem, in particular the Advantage form of the policy gradient which is detailed in [1]. We will add this clarification on our derivation as well.
> 3. In general if we have a double sum $\sum_{i=0} ( \sum_{j=i+1} a_i b_j )$, this is equivalent to $\sum_{j=0} \sum_{i<j} a_i b_j = \sum_{j=0} b_j \sum_{i<j} a_i$, this is because we are essentially summing over all tuples of natural numbers $(i,j)$ for which $j>i$. Now setting $a_i = \gamma^i A_i^i$ and $b_j = \gamma^{1+j} A^2_j \nabla_{\theta_1} \log \pi^1_j$ gives us the desired result.
> 4. We expand the steps as follows: $$
> \begin{aligned} \nabla_{\theta}V(\theta) &= E_{\tau \sim \pi} \sum_{t=0}^{\infty}\gamma^t A(a_t|s_t) \nabla_\theta \log \pi_{\theta}(a_t|s_t)\\\\
> &= \sum_{t=0}^{\infty} \gamma^t E_{\tau \sim \pi} [A(a_t|s_t) \nabla_\theta \log \pi_{\theta}(a_t|s_t)]\\\\
> &= \sum_{t=0}^{\infty} \gamma^t \sum_{s_t, a_t} P(a_t,s_t) A(a_t|s_t) \nabla_\theta \log \pi_{\theta}(a_t|s_t) \\\\
> &= \sum_{t=0}^{\infty} \gamma^t \sum_{s_t} P(s_t) \sum_{a_t} \pi_{\theta}(a_t|s_t) A(a_t|s_t) \nabla_\theta \log \pi_{\theta}(a_t|s_t) \\\\
> &= \sum_{t=0}^{\infty} \gamma^t \sum_s P(s_t=s) \sum_a \pi_{\theta}(a|s) A(a|s) \nabla_\theta \log \pi_{\theta}(a|s) \\\\
> &= \sum_s \left(\sum_{t=0}^{\infty} \gamma^t P(s_t=s)\right) \sum_a \pi_{\theta}(a|s) A(a|s) \nabla_\theta \log \pi_{\theta}(a|s) \\\\
> &= \sum_s d_{\gamma}(s) \sum_a \pi_{\theta}(a|s) A(a|s) \nabla_\theta \log \pi_{\theta}(a|s) \\\\
> &= \sum_{(s,a)} d_{\gamma}(s) \pi_{\theta}(a|s) A(a|s)\nabla_\theta \log \pi_\theta(a|s)
> \end{aligned}
> $$ Where from equation 2 to 3 we used the fact that for any function of a state-action pair \( f(a, s) \), we have: $$
> \begin{aligned}
> E_{\tau \sim \pi} f(a_t, s_t) &= \sum_\tau P(\tau) f(a_t, s_t) \\\\
> &= \sum_{a_t, s_t}  f(a_t, s_t) \sum_{t_1 \neq t} \sum_{t_2 \neq t} \dots P(\tau) \\\\
> &= \sum_{a_t, s_t} P(a_t, s_t) f(a_t, s_t)
> \end{aligned}
> $$where in the last step we have just marginalised $P(\tau)$ over all time-steps except for $t$, setting $f(a_t, s_t) = \nabla_\theta \log \pi_{\theta}(a_t|s_t)$ gives the intended result.
> 5. Assuming orthonormality simply means that $\nabla_{\theta} \log \pi_\theta(a|s) \cdot \nabla_{\theta} \log \pi_\theta(a'|s') = 0$ for all $(a,s) \neq (a', s')$, hence the sum over all state-action pairs collapses to a single term, which contains the $(a', s')$ pair. The gradient dot product for that pair is assumed to be $1$ again by the orthonormality assumption. We will note that this orthonormality assumption is exactly true in the tabular case. i.e. if the parameters $\theta_i$ of our policy are exactly $\log \pi(a|s)$ for all possible $(a,s)$, then this assumption is exact.
> 6. Here $A^*_{LOLA}(s_t, a_t, b_t)$ is the new advantage that should be put into the policy gradient theorem in order to get the update direction for a LOLA agent. It's not equal to $V^1$.
> 7. This is the assumption that LOLA makes about the opponent, it is true by hypothesis. LOLA takes $\Delta \theta_2 = \alpha\nabla_{\theta_2} V^2$ because it assumes that the opponent is a naive learner (i.e. doesn't incorporate opponent-shaping information), and hence would take a gradient step by greedily maximizing its value function.
> 8. Thank you, we will fix it.
> 9. In figure 1b, the state CD corresponds to a previous state which is "player 1 cooperated, and player 2 defected", DC is the reverse, "player 1 defected, and player 2 cooperated". This figure illustrates that our algorithm finds the tit-for-tat policy, which cooperates if the opponent cooperated in the last turn, but defects otherwise. This asymmetry between CD and DC is very important, cooperating in both states means that our agent doesn't punish opponent defection.
>
> We hope that our responses have satisfactorily addressed the reviewer's concerns, and we appreciate your positive comments regarding the originality, quality, clarity, and significance of our work. In this light we kindly ask the reviewer to reconsider their score.
>
> **References**
>
> [1] Agarwal, A., Jiang, N., Kakade, S., and Sun, W. (2021). Reinforcement Learning: Theory and Algorithms. Preprint.

---

> ### Comment · Reviewer_Niaw · 2024-11-21
>
> Thanks for your answers. Most of my concerns have been addressed. However, I suggest the authors may add the explanations for question 5 and question 3 to the revised paper.
>
> I will raise my score to 6, as I cannot recognize more impact of this paper that can convince me to give 8.

---

### Official Review · Reviewer_FsFC · 2024-11-02

**Soundness:** 3
**Presentation:** 4
**Contribution:** 3
**Rating:** 8
**Confidence:** 2

**Summary:**

This paper focuses on opponent shaping.  They propose an opponent shaping formulation that aligns the advantages of interacting agents, thereby simplifying the mathematical formulation of opponent shaping and reducing the associated computational complexity. In this formulation, the probability of future mutually beneficial actions is increased when their interaction has been positive. They demonstrate their methodologies' effectiveness on several social dilemmas and connect opponent shaping's past successes with this new advantage alignment formulation.

**Strengths:**

This paper is very well-written, and I appreciated how the authors helped the reader build intuition and understanding of the significance of their technique in section 4. The experiments served to reiterate the strength of their algorithm's performance, and the authors gave solid context for why each environment was selected.

**Weaknesses:**

The main concern I have with the paper is its individuality from the LOQA work, which concerns a similar technique, applied on similar problems, that achieves similar results. Both the more complex experiments (negociation and and harvest open) and the attached proofs in the appendix helped differentiate some of the beneficial aspects of Advantage Alignment in terms of scalability.

**Questions:**

- For the coin game, in the always defect case, LOQA receives slightly higher return than Advantage Alignment. Do you have any ideas for why this is?
-  In figure 3, there is mention of green values to show cooperation, but I do not see the green values in the figure.
- In Appendix 6, you mention that equation 59 uses the sum of past advantages of the agent up to the current time step and the advantage of the opponent at the current time step. Did you experiment with much tuning for number of terms to include in the sum of past advantages or number of future terms?

---

> ### Author Response · Authors · 2024-11-15
> **Addressing weaknesses and questions**
>
> We thank the reviewer for their kind comments and adress their concerns:
>
> **Weaknesses**
>
> The crucial difference between LOQA and Advantage Alignment lies in the ability to leverage PPO as a base optimizer instead of a Reinforce gradient estimator, which is very desirable for going beyond very simple toy environments like the coin game. We start from the same general assumptions as LOQA, but by phrasing our algorithm in terms of modifications to the advantages (and not through additional gradient terms like in LOQA), we can use off-the-shelf PPO. This greatly simplifies the implementation, and provides a much more scalable (in terms of environment complexity) opponent-shaping algorithm. LOQA by itself does not scale to environments like commons_harvest, so this is not a trivial difference, but a critical component needed for complex multi-agent tasks. LOQA also does not work for continuous action spaces, whereas Ad-Align is easily extendable to that case. Overall we see Ad-Align both as a more scalable successor to LOQA, and as a conceptual lens through which to view opponent-shaping as a whole, as mentioned in the response to all reviewers.
>
> **Questions:**
> 1. Our policies are neural network policies that condition on the entire history of observations, so it is very likely that the two algorithms converge to slightly different criterias for defection, i.e. how many times they observe defection before defecting themselves. On a related note, interpreting these policies is difficult for the exact same reason, the space of possible policies is very vast. What is interesting is that, although LOQA outperforms Advantage Alignment in the always-defect evaluation it does not in the evaluation against other exploitative policies like those of POLA and MFOS.
> 2. That is a typo, the caption should say that Blue values mean cooperation, thank you.
> 3. For the IPD, Coin Game and Negotiation Game experiments we used a context length of 16 and a discount factor of 0.9 without further experimentation. This is standard in the opponent-shaping community as it nicely approximates “infinite” returns. For the Meltingpot experiments we tried different context lengths of 30, 60, 80 and 100, with 30 working best. We believe the fundamental tradeoff between more expressive state representation and compute speed greatly favored speed in the context of Commons Harvest Open. In other words it is more valuable to have more parameter updates at the expense of context length in this environment.

---

### Official Review · Reviewer_GHM1 · 2024-11-04

**Soundness:** 4
**Presentation:** 4
**Contribution:** 2
**Rating:** 8
**Confidence:** 4

**Summary:**

This work aims to encourage cooperation in normal form games through opponent shaping. This is done by proposing Advantage Alignment, where the product of the advantages of the player's value-network and the opponent's value-network are used to update the policy. This means that, when advantages are aligned (i.e., they have the same sign), the log probabilities increase or decrease accordingly. The authors show theoretically that LOLA and LOQA, 2 other opponent shaping algorithms, also follow the advantage alignment principles. They then experimentally validate advantage alignment on Coin Game, Negotiation Game, and the high-dimensional Commons Harvest Open environment.

**Strengths:**

Strengths:
- This paper follows a long line of work on opponent shaping (LOLA, POLA, COLA, LOQA) and builds on LOQA to propose a new opponent shaping algorithm. While it feels a bit incremental, I find the work very well motivated, and theoretically justified. Extensive experiments on different domains and with relevant baselines confirm its practical performance.

**Weaknesses:**

Concerns:
- Having access to the opponent's value function results in a specific setting where all the player's preferences are public. The negotiation game completely changes if we know the preferences of the opponent, and we could devise a simple strategy that maximizes the average utility of both players. Since the utilities in this experiment are orthogonal to each other, there does not seem to be any real dilemma. For these reasons, I believe the insights gained from the Negotiation Game experiments to be limited.
- Advantage Alignment uses the product of advantages to align the policy. When there are more than 2 players involved (as in Commons Harvest Open), does the alignment depend on the product of the advantages of all players? If so, if a single agent does not cooperate, the whole alignment product fails. This would limit the scaling of Advantage Alignment to $n$ players.

**Questions:**

Additional clarification questions:
- Fig6 seems to indicate that Advantage Alignment is not really stable (increasingly noisy rewards, partial collapse towards the end of training against AC and AD). Is it sensible to hyperparameters? Or is there another reason for this behavior?
- Fig4 selects the best agent out of 10 seeds. Given the variance observed for the Negotiation Game in Fig6 (which granted is not the same game as Fig4), how representative are the results in Fig4?
- I am curious how Advantage Alignment can play against AD or AC if it can't change its policy (AD and AC are fixed policies). Doesn't this break Assumption 1?


Overall, I recommend for acceptance, for the strengths listed above (theoretically sound, well motivated, experimentally justified).

---

> ### Author Response · Authors · 2024-11-15
> **Addressing weaknesses and questions**
>
> We thank the reviewer for their positive comments and constructive feedback. We address the reviewer’s concerns and questions:
>
> **Concerns:**
>
> 1. First we want to clarify that preferences are not completely orthogonal, since the minimum value for any item is one, either player always gets utility for any extra item taken. We agree that the negotiation game changes if the preferences are public, and that a very simple strategy can maximize the utility of both players, but this makes the failure of naive PPO all the more evident! The environment should be trivial to solve, yet we see current methods fail even in this simplified case.
>
> 2. In the many-player scenario, the policy gradient term becomes:
> $$\nabla_{\theta_1}V^{1}(\mu) = E_{\tau}\left[\sum_{t=0}^\infty \gamma^t A^1(s_t, a_t, b_t) \left(\nabla_{\theta_1} \log \pi^1 (a_t | s_t) + \nabla_{\theta_1} \sum_{j\neq 1} \log \pi^j (b_t | s_t)\right)\right]$$
> And we will have a separate opponent-shaping term for each opponent (all of our math runs the same for each term in the sum over $j$). The new advantages do not depend on the product of all the players' advantages (which would indeed be catastrophic for scaling), but simply as the sum of 2-player advantages, i.e., our 2-player advantage update equation (for player 1, ommiting discount factors)
> $$A_t = A^1_t + \beta \left(\sum_{k<t} A^1_k \right)A^2_t$$
> becomes in the n-player case:
> $$A_t = A^1_t + \beta \left(\sum_{k<t} A^1_k\right)\left[\sum_{j\neq1}^n A^j_t\right] $$
> Which is simple to compute and has a natural interpretation as viewing the system of other $n-1$ agents as a single agent with advantages $\sum_{j\neq 1}^n A^j_t$. We believe this property of advantage alignment makes it especially suitable to many-player environments. It is also easy to implement using matrix multiplication.
>
> **Questions:**
>
>
> 1. This kind of instabilities are more apparent in the negotiation game but certainly not unique to it. Compared to single agent reinforcement learning, multi-agent reinforcement learning has the complication of the non-stationary property of the environment. Every time another agent updates their policy, this affects the transition dynamics, making training much more unstable. In our setup we have discovered that training against a (diverse) replay buffer of past policies is crucial for improving stability. However this solution is still not completely immune to complications. A common problem is that the replay buffer of agents saturates with policies with very similar behavior. In the case of the negotiation game we believe that the replay buffer saturates with cooperative policies and the agent forgets about retaliatory behavior since it is less likely to ever observe it in the first place. Ultimately this leads to performance drops against the Always Defect policy.
>
> 2. We want to establish that getting even a single seed that performs well in Meltingpot environments is already a very strong signal. The Meltingpot contest was won by single-seed hand-crafted policies that used domain knowledge about the environment to design strategies that maximize the utility of each individual game. Hence, getting a trained policy that performs well is already a non-trivial result. Having said that, out of our 10 seeds 3 get strong results temporarily at different stages of training. Since the stages of training are non-overlapping for the different seeds, it is hard to do an average-case analysis as it is standard in the single agent reinforcement learning literature. Since submitting, we have been arduously working on improving the stability of our runs by exploring auxiliary self-supervised losses and improving engineering components of our code (e.g. bigger agent replay buffers). Preliminary results look very promising, and our seeds are able to maintain high evaluation scores for significantly longer periods of time, compared to those at the time of submission.
>
> 3. This is related to our evaluation protocol (see reply to all reviewers), which is training our policies using the Advantage Alignment PPO policy gradient at training time and deploying in a zero-shot setting. As such, the assumptions are only valid at training time: since we train in a self-play setting, Assumption 1 holds. After training, we get a fixed policy that has dependence only on the past observations, therefore, it is straightforward to evaluate it against any opponent policy: we just run episodes against them.

---

> > ### Comment · Reviewer_GHM1 · 2024-11-21
> >
> > Thank you for your reply, and taking the time to answer all my concerns. Overall this clarifies the questions I had, especially concerning the evaluation protocol and scaling to $n$-player settings.

---

### Author Response · Authors · 2024-11-15
**Response to all reviewers**

We thank the reviewers for their feedback and helpful comments. We are very excited about the possible implications of this work and are happy to observe that the reviewers can appreciate this as well. Specifically, we are glad to hear that reviewers GHM1and V6em find our work very well motivated, and theoretically justified. Similarly we appreciate the reviewer's FsFC comment on our paper being very well-written. We now address some crucial points that were unclear and brought up independently by different groups of reviewers.

**Evaluation Protocol**

In the paper we forgot to share our evaluation protocol which is crucial to understand the setup that we are interested in. We thank reviewer V6em for pointing this out. In our evaluation protocol we train a policy using our algorithm, and deploy it in a zero-shot setting against a distribution of policies. We are interested in policies that can be deployed without fine-tuning and perform well at test time. As such we may do a stronger assumption than single-agent RL at training time: having access to the other agent’s rewards. This is a training harness that we get rid of at evaluation time, doesn’t affect the nature of the resulting policies (which don’t require access to other agent’s rewards) and ultimately leads to strong results.

**Significance of our contribution and differences with LOLA and LOQA**

We want to clarify the significance of our contribution and clarify the main differences with previous work. There are significant implementation differences between LOQA and advantage alignment. Ad-Align is fundamentally a modification to PPO: we use the same surrogate loss, but with modified advantage values. This allows us to obtain an algorithm with opponent-shaping effects while maintaining all the nice empirical properties of PPO. LOQA on the other hand follows a policy gradient without the PPO surrogate loss, which makes it less suitable to complex environments like the pixel-based commons-harvest. Moreover, LOQA’s normalization constant on the opponent’s policy assumption makes the algorithm intractable for continuous action spaces. Therefore, Advantage Alignment is a more applicable algorithm that demonstrates its advantages at a scale not seen before in the opponent-shaping literature, as shown in the Meltingpot 2.0. experiments.

There is also a more conceptual difference. We show that both LOQA and LOLA can be viewed as algorithms that implicitly do some form of advantage-alignment, i.e. including opponent-shaping effect into a learning algorithm seems intimately tied to including terms of the form $A^1_t A^2_k$ in the advantages. This relationship is obscured when looking at LOQA and LOLA in their published forms. As such, Advantage Alignment provides a more general framework in which to see future multi-agent policy gradient algorithms.

This makes Advantage-Alignment both a better starting point for future progress in opponent-shaping, and a unifying framework for past algorithms. We agree that this difference to past algorithms was not properly communicated in the paper, and we thank the reviewers for their comments. We will update the paper to more clearly articulate these points.

---

### Author Response · Authors · 2024-11-21
**Updated paper revision**

Dear Reviewers and Area Chairs,

We have submitted a revision to the paper adding the evaluation protocol, writing the extension to $n$-player games and fixing multiple typos. The changes in the new submission are highlighted in **green**. Specifically we addressed the following concerns/typos brought up by the following reviewers:

**Reviewer Niaw**:
1.  Add missing $\beta$ factor in the Advantage Alignment derivation. (Page 14, lines 719-755)
2.  Add details on how equation (16) is derived from equation (15). (Page 14, line 749)
3.  Remove the "equation equation" typo in line 1001. (Page 19, line 1001)

**Reviewer V6em**:
1. Add description of our evaluation protocol in the Experiments section. (Page 6, lines 290-291)
2. Add definition of the Advantage function. (Page 3, line 133)
3. Fix typo in equation (6), replacing $\pi^2$ by $\hat{\pi}^2$. (Page 3, line 156)

**Reviewer GHM1**:
1. Add an appendix section on $n$-player Advantage Alignment. (Page 20, line 1066)

**Reviewer FsFC**:
1. Fix typo in Figure 3, replacing "green" by "blue". (Page 8, line 402)

We are eager to engage in further discussion and would greatly appreciate any additional feedback or insights, especially from reviewers who have not yet had the opportunity to share their thoughts at the time of this revision (Niaw, FsFC).

---

### Meta-Review · Area_Chair_ChFE · 2024-12-20

**Metareview:**

This paper proposes a new opponent shaping algorithm and a unified interpretation encompassing several others like LOLA and its relatives. They tested their algorithm on various environments including challenging ones such as Melting Pot environments. After discussion, all reviewers were happy with the paper.

**Additional Comments On Reviewer Discussion:**

Reviewers were happy with the paper after discussion and one raised their score after reading the author's response.

---

### Decision · Program_Chairs · 2025-01-22

Accept (Oral)